# Inflammasome activation leads to cDC1-independent cross-priming of CD8 T cells by epithelial cell-derived antigen

Katherine A Deets[1], Randilea Nichols Doyle[1†], Isabella Rauch[2], Russell E Vance[1,3,4*]

[1]Division of Immunology and Pathogenesis, Department of Molecular and Cell Biology, University of California, Berkeley, Berkeley, United States; [2]Department of Molecular Microbiology and Immunology, Oregon Health and Science University, Portland, United States; [3]Cancer Research Laboratory, University of California, Berkeley, Berkeley, United States; [4]Howard Hughes Medical Institute, University of California, Berkeley, Berkeley, United States

**\*For correspondence:**
rvance@berkeley.edu

**Present address:** [†]Department of Microbiology, Immunology and Molecular Genetics, David Geffen School of Medicine, University of California, Los Angeles, Los Angeles, United States

**Abstract** The innate immune system detects pathogens and initiates adaptive immune responses. Inflammasomes are central components of the innate immune system, but whether inflammasomes provide sufficient signals to activate adaptive immunity is unclear. In intestinal epithelial cells (IECs), inflammasomes activate a lytic form of cell death called pyroptosis, leading to epithelial cell expulsion and the release of cytokines. Here, we employed a genetic system to show that simultaneous antigen expression and inflammasome activation specifically in IECs is sufficient to activate CD8[+] T cells. By genetic elimination of direct T cell priming by IECs, we found that IEC-derived antigens were cross-presented to CD8[+] T cells. However, cross-presentation of IEC-derived antigen to CD8[+] T cells only partially depended on IEC pyroptosis. In the absence of inflammasome activation, cross-priming of CD8[+] T cells required *Batf3*[+] dendritic cells (conventional type one dendritic cells [cDC1]), whereas cross-priming in the presence of inflammasome activation required a *Zbtb46*[+] but *Batf3*-independent cDC population. These data suggest the existence of parallel inflammasome-dependent and inflammasome-independent pathways for cross-presentation of IEC-derived antigens.

## Editor's evaluation

This study uses sophisticated genetic tools to demonstrate that intestinal epithelial-derived antigens can be cross-presented by dendritic cells to activate CD8[+] T cells via pyroptosis-dependent and -independent pathways. The study provides novel insight into how inflammasome activation regulates CD8 T cell responses. This paper will be of interest to scientists that are interested in both innate and adaptive immune systems.

## Introduction

The innate immune system provides a crucial first line of defense against invading pathogens, and in addition, activates and guides subsequent adaptive immune responses. Although the role of innate immunity in promoting adaptive immunity has long been appreciated (*Janeway, 1989*), most studies have focused on the contributions of Toll-like receptors (TLRs), and significantly less is known about how other innate immune pathways influence the adaptive immune system (*McDaniel et al., 2021*).

Inflammasomes are a heterogeneous group of cytosolic innate immune sensors, each of which initiates signaling in response to specific stimuli, including pathogen-associated molecules and activities

or cellular damage (*Awad et al., 2018*; *Downs et al., 2020*; *Palazon-Riquelme and Lopez-Castejon, 2018*). Regardless of the input signal, a common output of inflammasome activation is the recruitment and activation of caspase proteases (e.g., Caspase-1), which cleave and activate the inflammatory cytokines pro-interleukin (IL)-1β and pro-IL-18 and/or the pore-forming protein gasdermin D. Active gasdermin D oligomerizes in the plasma membrane to form pores that serve as a conduit for the release of active IL-1β and IL-18. Active gasdermin D can also initiate pyroptotic cell death and/or lysis (*de Vasconcelos et al., 2019*; *DiPeso et al., 2017*; *Evavold et al., 2018*; *He et al., 2015*; *Heilig et al., 2018*; *Kayagaki et al., 2015*; *Shi et al., 2015*). In intestinal epithelial cells (IECs), inflammasome activation also results in the expulsion of cells from the epithelial monolayer into the intestinal lumen. Pyroptosis and cell expulsion provide host defense against intracellular pathogens by eliminating their replicative niche (*Fattinger et al., 2021*; *Hausmann et al., 2020*; *Mitchell et al., 2020*; *Rauch et al., 2017*; *Sellin et al., 2014*).

The role of inflammasome activation during adaptive immunity remains incompletely understood, and inflammasomes appear to have both beneficial and detrimental effects on the adaptive response, depending on the context (*Deets and Vance, 2021*; *Evavold and Kagan, 2018*). In addition, most studies to date use whole-animal knockouts and intravenous infection models, making it difficult to draw conclusions about the effects of inflammasome activation within different cell types. While there remains limited evidence on how inflammasome activation and pyroptosis of either hematopoietic cells or IECs impacts presentation of cell-derived antigens, systemic IL-1β and IL-18 have been implicated in driving type one helper T cell (Th1), Th17, and CD8$^+$ T cell immunity following bacterial infections (*Kupz et al., 2012*; *O'Donnell et al., 2014*; *Pham et al., 2017*; *Tourlomousis et al., 2020*; *Trunk and Oxenius, 2012*). Likewise, CD4$^+$ and CD8$^+$ T cell responses to influenza have been shown to require inflammasome signaling components, presumably in lung macrophages (*Ichinohe et al., 2009*). However, inflammasome activation has also been found to inhibit T cell-mediated immunity (*Sauer et al., 2011*; *Theisen and Sauer, 2017*), including through the pyroptotic destruction of key antigen presenting cells (APCs) (*McDaniel et al., 2020*; *Tourlomousis et al., 2020*). Inflammasomes have also been suggested to influence adaptive responses to tumors (reviewed in *Deets and Vance, 2021*; *Evavold and Kagan, 2018*).

Most studies to date have evaluated the effects of inflammasome activation on adaptive immunity in the context of infections. Though physiologically relevant, microbial infections are also complex to analyze since they engage multiple innate receptors, including TLRs. It thus remains unknown whether inflammasome activation alone provides sufficient co-stimulatory signals to initiate an adaptive response, and if so, which inflammasome-containing cell populations can drive this response. In addition, the fate of antigens after inflammasome activation remains poorly understood. Conceivably, expulsion of pyroptotic epithelial cells may result in the loss of antigen, thereby hindering adaptive immunity, or alternatively, pyroptosis may promote the release of epithelial or hematopoietic cell antigens to APCs to activate adaptive immunity.

To investigate how inflammasome activation might influence adaptive immunity, we focused on the NAIP–NLRC4 inflammasomes, which specifically respond to flagellin (via NAIP5/6) or bacterial type III secretion system proteins (via NAIP1/2) (*Kofoed and Vance, 2011*; *Rauch et al., 2016*; *Zhao et al., 2016*; *Zhao et al., 2011*). Although most inflammasomes require the adaptor protein Apoptosis-associated Speck-like protein containing a Caspase activation and recruitment domain (ASC) to recruit and activate pro-Caspase-1, NLRC4 is able to bind and activate pro-Caspase-1 directly—though ASC (encoded by the *Pycard* gene) has been found to enhance the production of IL-1β and IL-18 (*Broz et al., 2010a*; *Broz et al., 2010b*; *Mariathasan et al., 2004*). Cleavage of gasdermin D does not require ASC following NAIP–NLRC4 activation (*He et al., 2015*; *Kayagaki et al., 2015*; *Shi et al., 2015*). NAIPs and NLRC4 are highly expressed in IECs, where they provide defense against enteric bacterial pathogens including *Citrobacter* (*Nordlander et al., 2014*), *Salmonella* (*Fattinger et al., 2021*; *Hausmann et al., 2020*; *Rauch et al., 2017*; *Sellin et al., 2014*), and *Shigella* (*Mitchell et al., 2020*). Inflammasome-driven IEC expulsion appears to be a major mechanism by which NAIP–NLRC4 provides innate defense against enteric pathogens. However, it is not currently known how pyroptosis and IEC expulsion influence the availability of IEC-derived antigens and what impact this has on the adaptive immune response.

Indeed, even at steady state in the absence of inflammasome activation and pyroptosis, it remains unclear how antigens present in IECs are delivered to APCs to stimulate adaptive immune responses,

or whether perhaps IECs can directly activate T cells (*Chulkina et al., 2020*; *Heuberger et al., 2021*; *Liu and Lefrançois, 2004*; *Nakazawa et al., 2004*). Conventional type one dendritic cells (cDC1s) are thought to acquire apoptotic bodies from IECs and shuttle the cell-associated antigens through the MHC II pathway to drive a tolerogenic CD4[+] T cell response under homeostatic conditions (*Cummings et al., 2016*; *Huang et al., 2000*). cDC1s were also recently shown to induce Foxp3[+]CD8[+] $T_{regs}$ through the cross-presentation of IEC-derived tissue-specific antigens (*Joeris et al., 2021*). Additionally, in the context of inflammation, a subset of migratory cDC1s have been shown to also take up IEC-derived antigen to activate CD8[+] T cells; however, it remains unclear how these cDC1s acquire IEC-derived antigen.

Because NAIP–NLRC4 activation can result in IEC pyroptosis prior to the expulsion of IECs from the epithelium (*Rauch et al., 2017*), we hypothesized that cell lysis could release antigen basolaterally, which could then be taken up by cDC1s and cross-presented to CD8[+] T cells. To address the role of inflammasome-induced cell death in antigen presentation and subsequent activation of CD8[+] T cells, we used a genetic mouse model in which an ovalbumin (Ova)-flagellin (Fla) fusion protein is inducibly expressed specifically in IECs (*Nichols et al., 2017b*). The OvaFla fusion protein provides a model antigenic epitope (SIINFEKL) to activate specific CD8[+] (OT-I) T cells (*Hogquist et al., 1994*), concomitant with the activation of the NAIP–NLRC4 inflammasome by a C-terminal fragment of flagellin that does not activate TLR5. This genetic system has the advantage of selectively activating inflammasome responses in the absence of exogenous or pathogen-derived TLR ligands, allowing us to address the sufficiency of inflammasome activation for adaptive responses. Our results suggest the existence of distinct NLRC4-dependent and NLRC4-independent pathways for cross-presentation of IEC-derived antigens in vivo.

## Results

### Genetic system for NAIP–NLRC4 activation in IECs

We took advantage of a previously established mouse model (*Nichols et al., 2017b*) that allows for Cre-inducible and cell type-specific NAIP–NLRC4 activation (*Figure 1A*). These mice harbor an OvaFla gene fusion that encodes a non-secreted chicken ovalbumin protein—a model antigen—fused to the C-terminal 166 amino acids of flagellin that functions as an agonist of NAIP–NLRC4 but not TLR5 (*Nichols et al., 2017b*). The OvaFla gene is inserted within the constitutively expressed *Rosa26* locus, downstream of a floxed transcriptional stop cassette and upstream of an IRES-GFP cassette. To create a genetic system for inducible NAIP–NLRC4 activation in IECs, we crossed the OvaFla mice to Villin-Cre-ER[T2] mice (*el Marjou et al., 2004*), which harbor a tamoxifen-inducible Cre recombinase driven by the *Villin* promoter. The resulting OvaFla Villin-Cre-ER[T2] (hereafter shortened to 'OvaFla') mice respond to tamoxifen administration by expressing Cre, and subsequently the OvaFla protein, specifically in IECs. To study the influence of NAIP–NLRC4 activation, pyroptosis, and cytokine production on CD8[+] T cell activation, we generated *Nlrc4*[−/−], *Gsdmd*[−/−], and *Pycard*[−/−] OvaFla lines.

Tamoxifen is typically administered in a corn oil emulsion through oral gavage or intraperitoneal injection. In preliminary experiments, we found corn oil contains trace bacterial contaminants that activate TLR signaling (*Nichols, 2017a*). Thus, to avoid confounding effects of TLR activation, and to isolate the specific effects of inflammasome activation, we administered tamoxifen orally through a commercially available tamoxifen-containing chow. OvaFla mice were fed ab libitum, and their weight and temperature were tracked daily as previously described indicators of NAIP–NLRC4 activation (*von Moltke et al., 2012*). After a single day on the tamoxifen diet, wild-type (WT) OvaFla and *Gsdmd*[−/−] OvaFla mice lost a significant amount of weight, and by day 2 of the tamoxifen diet, these mice exceeded the humane weight loss endpoint on our animal protocol and were euthanized (*Figure 1B*, top). In contrast, the *Nlrc4*[−/−] OvaFla mice, as well as the OvaFla-only and Cre-only littermate control mice, maintained a consistent body weight and appeared healthy over the 2-day time course. Although not statistically significant, the WT OvaFla and *Gsdmd*[−/−] OvaFla mice also exhibited decreases in core body temperature by day 2 relative to the *Nlrc4*[−/−] OvaFla mice (*Figure 1B*, bottom), consistent with previous analyses using recombinant flagellin protein (FlaTox) to induce acute NAIP–NLRC4 activation (*Rauch et al., 2017*; *von Moltke et al., 2012*).

Serum was collected from OvaFla mice at day 2 of the tamoxifen diet and assayed for IL-18, which is released from IECs following NAIP–NLRC4 activation (*Rauch et al., 2017*). The serum of WT OvaFla

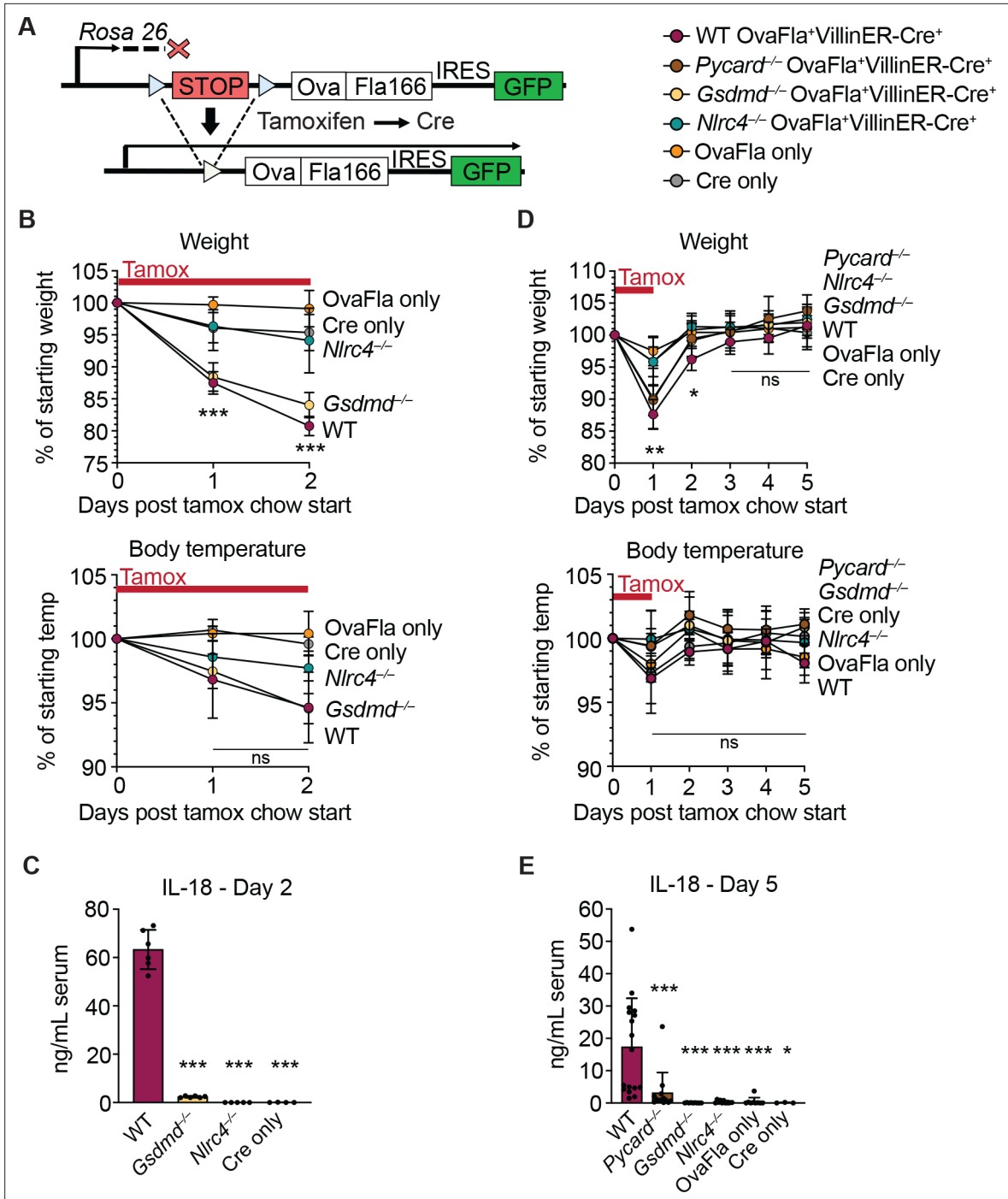

**Figure 1.** Genetic OvaFla VillinER-Cre^T2 system results in NAIP–NLRC4 activation in intestinal epithelial cells (IECs) of mice upon tamoxifen chow administration. (**A**) Schematic of the OvaFla gene cassette in the *Rosa26* locus. The cassette contains full-length mature (non-secreted) ovalbumin fused to the C-terminal 166 amino acids of flagellin, and an IRES-GFP. When OvaFla mice are crossed to mice containing the tamoxifen-inducible Villin-ER-Cre^T2, tamoxifen administration results in Cre-controlled excision of the stop cassette and expression of the OvaFla fusion protein and GFP within IECs. (**B**) Daily weight (top) and rectal temperature (bottom) measurements of OvaFla mice during a 2-day course of tamoxifen chow (depicted as red bar). (**C**) Quantification of IL-18 ELISA performed on serum from the mice shown in panel B at day 2 post tamoxifen chow start. Each dot represents an individual mouse. (**D**) Daily weight (top) and rectal temperature (bottom) measurements of OvaFla mice following a single day pulse of tamoxifen chow (depicted as red bar). (**E**) Quantification of IL-18 ELISA performed on serum from the mice shown in panel D at day 5 post tamoxifen chow start. (**B–E**) Data shown as mean ± SD and are from a single representative experiment. Each dot represents an individual mouse. Significance calculated using one-way ANOVA and Tukey's multiple comparisons test (*p < 0.05, **p < 0.01, ***p < 0.001). See *Figure 1—source data 1* for exact p values.

*Figure 1 continued on next page*

*Figure 1 continued*

The online version of this article includes the following source data for figure 1:

**Source data 1.** Statistical data and individual data points for *Figure 1*.

mice contained approximately 60 times more IL-18 than the serum of *Gsdmd*$^{-/-}$ OvaFla mice, demonstrating that gasdermin D is required for IL-18 release from IECs following NAIP–NLRC4 activation (*Figure 1C*). IL-18 was not detected in the *Nlrc4*$^{-/-}$ mice or in the OvaFla-only or Cre-only littermate controls. Taken together, these data show that the OvaFla system results in robust NAIP–NLRC4 activation in IECs following tamoxifen administration.

To limit confounding effects of morbidity in the NAIP–NLRC4 sufficient strains, we shortened the administration of tamoxifen chow to a single day pulse. We again monitored weight and rectal temperature each day. We found that while the WT, *Pycard*$^{-/-}$, and *Gsdmd*$^{-/-}$ OvaFla mice initially lost weight, weight loss was reversed within 2 days of being fed normal chow (*Figure 1D*, top). No significant difference in core body temperature was found between strains over the 5-day experiment (*Figure 1D*, bottom).

Serum was collected at day 5 post start of the tamoxifen chow diet and again assayed for IL-18 through ELISA. Similar to the 2-day tamoxifen pulse, a single day of tamoxifen chow resulted in significant IL-18 production in the WT OvaFla mice but minimal to no detectable IL-18 in the other OvaFla strains (*Figure 1E*). The WT mice exhibited heterogeneity in the IL-18 response with the single day chow pulse, which may be related to some mice being averse to consuming the tamoxifen chow (*Chiang et al., 2010*) or heterogeneity in the kinetics of the response.

We also performed immunofluorescence imaging of the small intestines of mice from each of the OvaFla lines after a single day pulse of tamoxifen chow. The presence of an IRES-GFP downstream of the OvaFla gene allows us to track the expression of the transgene. While approximately 30% of the IECs were GFP$^+$ in *Nlrc4*$^{-/-}$ OvaFla mice, only about 2% of the IECs were GFP$^+$ in the WT, *Pycard*$^{-/-}$, or *Gsdmd*$^{-/-}$ OvaFla mice at that time point (*Figure 2A and B*). Additionally, of those GFP$^+$ cells, IECs in the *Nlrc4*$^{-/-}$ OvaFla mice contained significantly more GFP signal when compared with the other OvaFla lines, whereas transgene expression was indistinguishable among WT, *Pycard*$^{-/-}$, and *Gsdmd*$^{-/-}$ mice (*Figure 2C*). Low transgene expression in genotypes other than *Nlrc4*$^{-/-}$ was anticipated because previous work (*Rauch et al., 2017*; *Sellin et al., 2014*) found that IECs are rapidly expelled from the epithelium upon NAIP–NLRC4 activation. Given that we observe robust IL-18 levels in the serum of WT mice (*Figure 1C and E*), we believe the transgene is expressed in WT (and *Pycard*$^{-/-}$ and *Gsdmd*$^{-/-}$) mice, but NLRC4$^+$ cells that express high levels of the transgene are expelled, limiting our ability to detect them. Although pyroptosis of IECs requires gasdermin D, NAIP–NLRC4-induced IEC expulsion was previously found to be independent of gasdermin D, likely due to the existence of an NLRC4-Caspase-8-dependent apoptosis pathway that also leads to IEC expulsion (*Man et al., 2013*; *Rauch et al., 2017*).

Taken together, these data show that OvaFla production under control of the tamoxifen-inducible Villin-Cre-ER$^{T2}$ system results in robust NAIP–NLRC4 activation in the IECs of mice. A single day pulse of tamoxifen chow leads to significant IL-18 production without gross morbidity or mortality in the NAIP–NLRC4 sufficient strains. Additionally, OvaFla likely accumulates in the IECs of the *Nlrc4*$^{-/-}$ OvaFla mice, as these cells do not undergo NAIP–NLRC4-driven cell expulsion.

## CD8$^+$ T cell activation by epithelial antigens

To understand how NAIP–NLRC4 activation influences IEC-derived antigen release and presentation, we followed the response of Ova-specific TCR transgenic OT-I CD8$^+$ T cells following OvaFla induction in each of our mouse lines. Congenically marked (CD45.1$^+$ or CD45.1$^+$ CD45.2$^+$) OT-I T cells were harvested from the spleens and mesenteric lymph nodes of OT-I *Rag2*$^{-/-}$ mice, labeled with CellTrace Violet proliferation dye, and intravenously transferred into the OvaFla mice ($2 \times 10^4$ cells per mouse) (*Figure 3A*). Immediately following adoptive transfer, the mice were placed on tamoxifen chow for a single day. At day 5 post adoptive transfer, the mice were euthanized, and their mesenteric lymph nodes, which drain immune cells from the intestines (*Esterházy et al., 2019*), and spleens were analyzed for OT-I T cell proliferation and activation.

A dividing OT-I population was identified by flow cytometry in each Cre$^+$ OvaFla line (*Figure 3—figure supplement 1A and B*), indicating that antigens expressed in IECs can be processed and

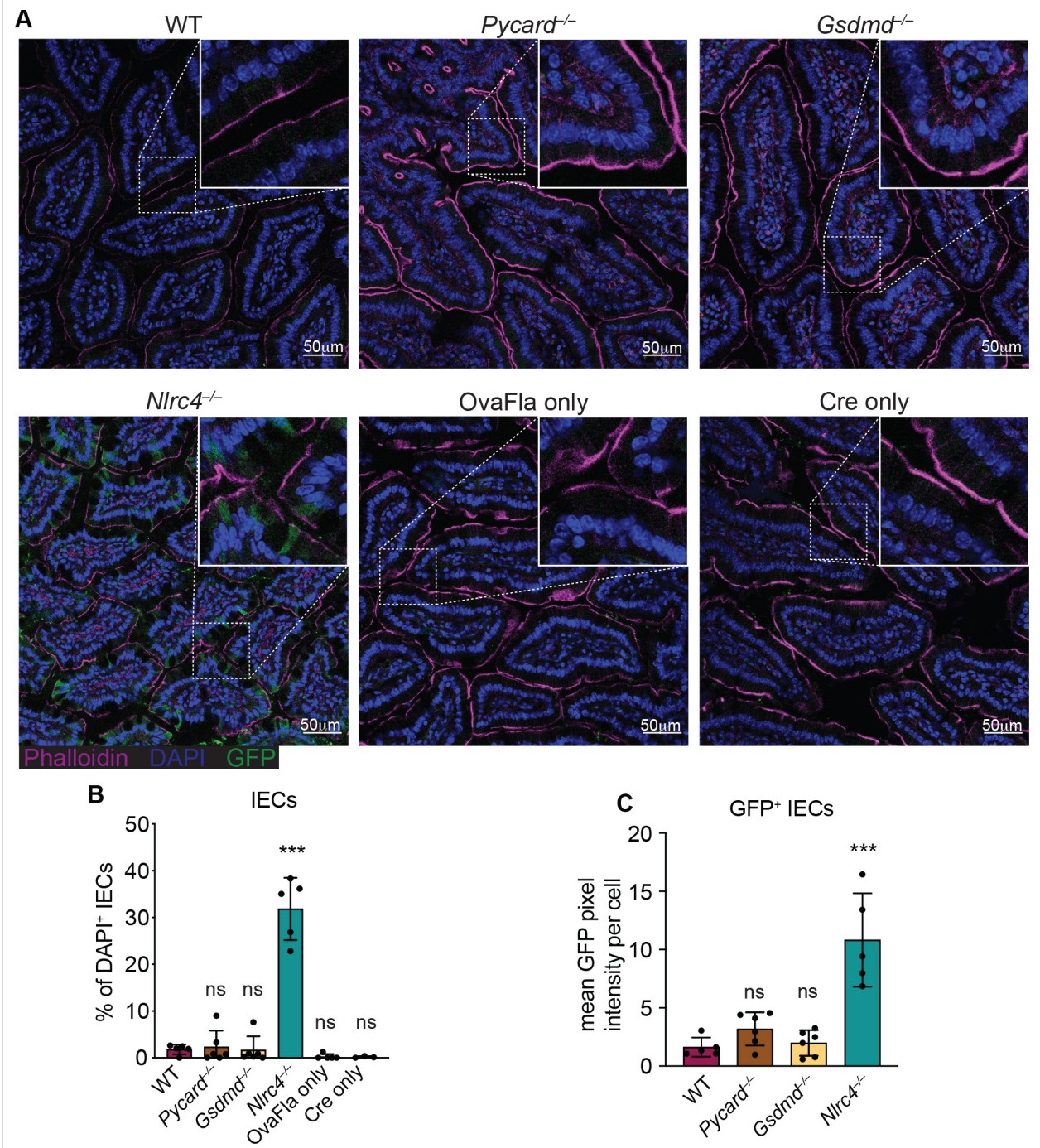

**Figure 2.** GFP⁺ cells accumulate in *Nlrc4⁻/⁻* OvaFla mice following tamoxifen administration. (**A**) Representative immunofluorescence images of the small intestines of indicated OvaFla mice on day 2 following a single day pulse of tamoxifen chow. (**B**) Quantification of DAPI⁺ IECs that are also GFP⁺ for each OvaFla line. Approximately 100 cells from least 15 separate villi across four to five images were counted per mouse. (**C**) Quantification of mean GFP pixel intensity for GFP⁺ intestinal epithelial cells (IECs) in each OvaFla line. Data represent an averaged value from 12 to 20 cells per image across four to five images per mouse. (**B–C**) Data are pooled from two biological replicates, and each dot represents an individual mouse. Data shown as mean ± SD. Significance calculated using one-way ANOVA and Tukey's multiple comparisons test (*p < 0.05, **p < 0.01, ***p < 0.001). Only p values between wild-type (WT) and other experimental groups are shown. See *Figure 2—source data 1* for exact p values.

The online version of this article includes the following source data for figure 2:

**Source data 1.** Statistical data for *Figure 2*.

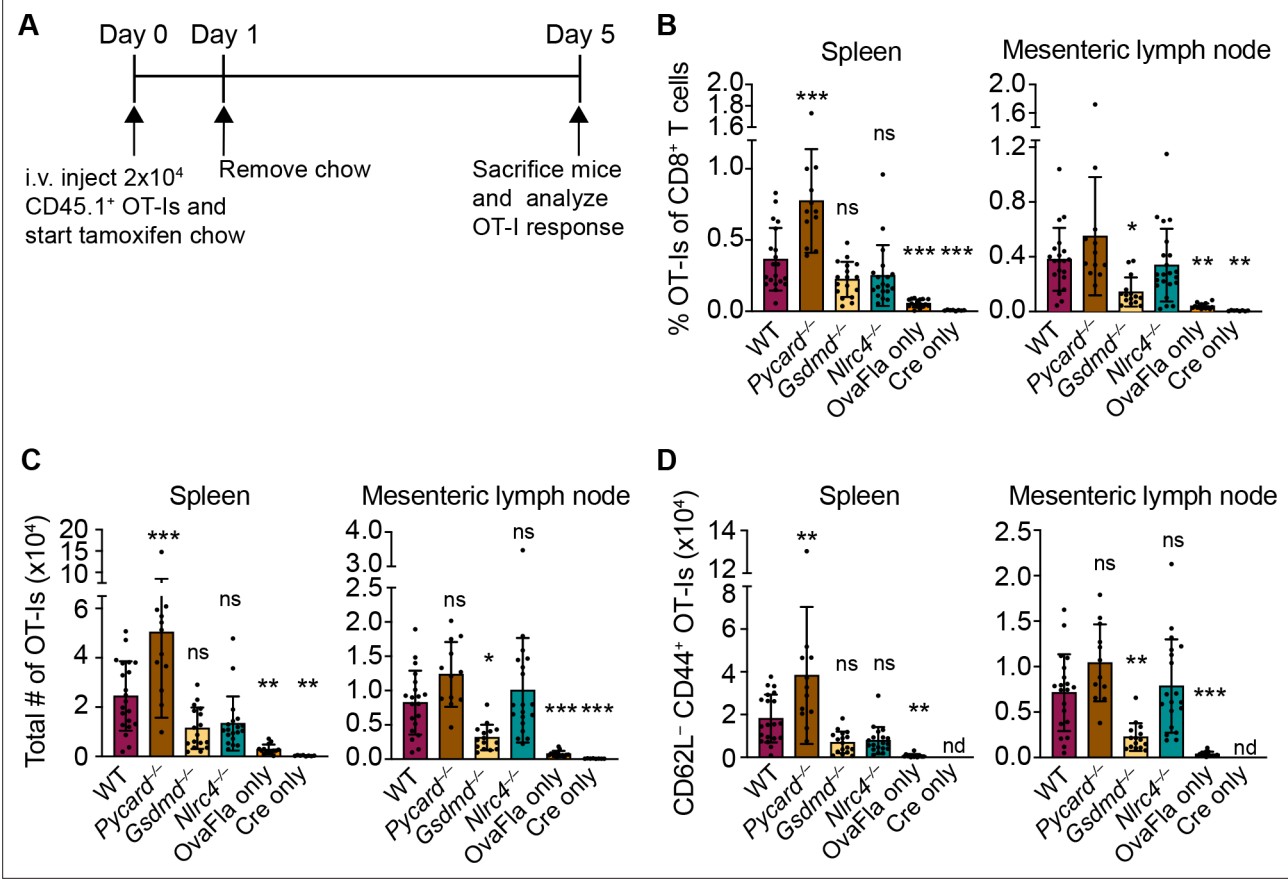

**Figure 3.** OvaFla expression in intestinal epithelial cells (IECs) results in OT-I proliferation and activation that is independent of ASC and NLRC4 but partially dependent on gasdermin D. (**A**) Overview of experimental setup for analyzing OT-I responses to OvaFla production in IECs. (**B**) Quantification of OT-Is as a percent of total CD8+ T cells per spleen (left) and mesenteric lymph node (right). (**C**) Total number of OT-Is per spleen (left) and mesenteric lymph node (right). (**D**) Total number of CD62L−CD44+ OT-Is per spleen (left) and mesenteric lymph node (right). Samples with fewer than 20 OT-Is were excluded from CD62L, CD44 calculations. Tissues were harvested and analyzed at day 5 post tamoxifen chow start. Data are pooled from three biological replicates, and each dot represents an individual mouse. Data shown as mean ± SD. Significance calculated using one-way ANOVA and Tukey's multiple comparisons test (*p < 0.05, **p < 0.01, ***p < 0.001). Only p values between wild-type (WT) and other experimental groups are shown. See *Figure 3—source data 1* for exact p values.

The online version of this article includes the following source data and figure supplement(s) for figure 3:

**Source data 1.** Statistical data for *Figure 3*.

**Figure supplement 1.** OvaFla expression in intestinal epithelial cells (IECs) results in OT-I proliferation and activation that is independent of ASC and NLRC4 but partially dependent on gasdermin D.

**Figure supplement 1—source data 1.** Statistical data for *Figure 3—figure supplement 1*.

presented to activate CD8+ T cells in vivo. Surprisingly, however, there was minimal difference in the relative percent (*Figure 3B*), absolute number (*Figure 3C*), or activation status (defined as CD62L−CD44+) (*Figure 3D*, *Figure 3—figure supplement 1C,D*) of OT-I T cells between the WT and *Nlrc4−/−* OvaFla mice in either the spleen or mesenteric lymph node. In fact, relative to the WT OvaFla mice, a higher percent of the OT-I T cells in the *Nlrc4−/−* OvaFla mice produced IFNγ and TNFα following ex vivo stimulation with PMA and ionomycin (*Figure 3—figure supplement 1E*). These data indicate OT-I T cells respond to IEC-expressed Ova in a manner that is independent of NAIP–NLRC4 activation. However, the specific lack of IEC expulsion and the resulting higher accumulation of antigen in IECs in *Nlrc4−/−* mice (*Figure 2*) means that the WT and *Nlrc4−/−* mice are not truly comparable.

In contrast to *Nlrc4−/−* IECs, both *Pycard−/−* and *Gsdmd−/−* IECs are expelled after inflammasome activation and thus exhibit indistinguishably low OvaFla-IRES-GFP transgene expression in IECs as compared to WT mice (*Figure 2*). Both strains are also defective for IL-18 release (*Figure 1C and E*). The major difference between the two strains is that *Pycard−/−* cells can still undergo *Gsdmd*-dependent

pyroptosis, whereas $Gsdmd^{-/-}$ cells do not undergo lytic pyroptosis but are nevertheless expelled from the epithelium as intact apoptotic cells, likely via a Caspase-1 and/or -8 pathway (*Man et al., 2013*; *Rauch et al., 2017*). There was little difference in OT-I numbers (*Figure 3B*, right, *Figure 3C*, right) or activation (*Figure 3D*, right, *Figure 3—figure supplement 1C,D*, right), as well as no difference in OT-I IFNγ and TNFα production (*Figure 3—figure supplement 1E*), in the mesenteric lymph nodes of the WT versus $Pycard^{-/-}$ OvaFla mice. However, there were significantly more activated OT-Is in the spleens of the $Pycard^{-/-}$ OvaFla mice (*Figure 3C*, right, *Figure 3D*, right). These data suggest there may be some suppressive role for ASC in NAIP–NLRC4-dependent activation of CD8+ T cells in circulation, though future characterization of these findings is needed. In contrast to the $Pycard^{-/-}$ OvaFla mice, the $Gsdmd^{-/-}$ OvaFla mice had a significantly lower number of activated cells relative to the WT OvaFla mice, but this difference was only found in the mesenteric lymph nodes (*Figure 3C and D*, *Figure 3—figure supplement 1D*). Taken together, these results suggest that inflammasome activation in IECs is not essential for OT-I CD8+ T cell activation, yet gasdermin D-mediated pyroptosis of IECs may play a partial role (see Discussion).

We also observed that a small percentage of OT-Is in the OvaFla only (Cre-minus) mice appear to be activated (*Figure 3—figure supplement 1D, E*). Since these mice are lacking Cre recombinase, we suspect there may be a very low level of Cre-independent expression of the OvaFla transgene. This chronic OvaFla expression is likely to result in exhaustion and/or deletion of any endogenous Ova-specific effector T cells (*Kurachi, 2019*). Indeed, we were unable to identify any SIINFEKL-specific endogenous CD8+ T cells via tetramer staining or ELISpot assays. Furthermore, tamoxifen-induced estrogen receptor signaling in the Villin-Cre-ER^T2 mice is known to occur in crypt stem cells, which leads to tamoxifen-independent Cre expression in the IEC progeny (*el Marjou et al., 2004*). Tamoxifen-independent Cre expression in the OvaFla mice could cause OvaFla to become a chronic stimulus, again likely leading to CD8+ T cell exhaustion. Because of these potentially complicating factors, we believe our OvaFla system is best suited to follow the immediate fate of IEC-derived antigen using naïve transferred OT-I transgenic T cells.

## Cross-presentation of IEC antigens

IECs express MHC class I on their surface and are capable of directly presenting antigen to CD8+ T cells (*Christ and Blumberg, 1997*; *Nakazawa et al., 2004*). It is therefore possible that the OT-I activation seen in the OvaFla mice is a result of direct presentation of Ova peptide by the IECs expressing OvaFla. However, it is also possible that the OT-I T cells are being cross-primed by cDC1s that engulf and 'cross-present' the IEC-derived Ova (*Cerovic et al., 2015*; *Liu and Lefrançois, 2004*). The fate of IEC-derived antigens and the role of antigen presentation pathways leading to CD8+ T cell activation has not previously been addressed with a completely in vivo system that can genetically distinguish cross from direct presentation of IEC antigens.

To determine whether the OT-Is are being activated through cross-presentation or direct presentation of Ova peptide, we took advantage of the H-2K^bm1 mouse model that contains a seven base pair mutation in the gene encoding K^b (*Schulze et al., 1983*). The bm1 mutation renders K^b unable to bind the Ova-derived OT-I agonist peptide, SIINFEKL (*Nikolić-Žugić and Bevan, 1990*). We bred H-2K^bm1 mice to each of our OvaFla lines to establish mice that make OvaFla in their IECs but are incapable of directly presenting the SIINFEKL peptide (H-2K^bm1+ OvaFla mice, referred to here as bm1+ OvaFla mice). We then generated bone marrow chimeras using bm1+ OvaFla mice as lethally irradiated recipients that were reconstituted with WT H-2K^b bone marrow from B6 CD45.1 donors (*Figure 4A*, left). In these chimeras, the IECs produce OvaFla following tamoxifen administration, but the IECs themselves are unable to present SIINFEKL to OT-I T cells. The donor-derived hematopoietic cells, including cross-presenting cDC1s, do not contain the OvaFla gene cassette but are able to cross-present the SIINFEKL peptide if they acquire it from IECs (*Figure 4A*, right). Therefore, in the bm1+ OvaFla chimeras, OT-I proliferation and activation will only be observed if the SIINFEKL peptide is cross-presented.

Eight to ten weeks after lethal irradiation and reconstitution, bm1+ OvaFla mice received $2 \times 10^4$ CD45.1+ CD45.2+ CellTrace Violet labeled OT-I T cells intravenously and were given a 1-day pulse of tamoxifen chow (*Figure 4A*, left). The mice were euthanized at day 5 post OT-I transfer, and their spleens and mesenteric lymph nodes were analyzed for OT-I proliferation and activation. Serum was also collected for IL-18 ELISA to confirm NAIP–NLRC4-dependent IL-18 release following OvaFla induction (*Figure 4—figure supplement 1A*).

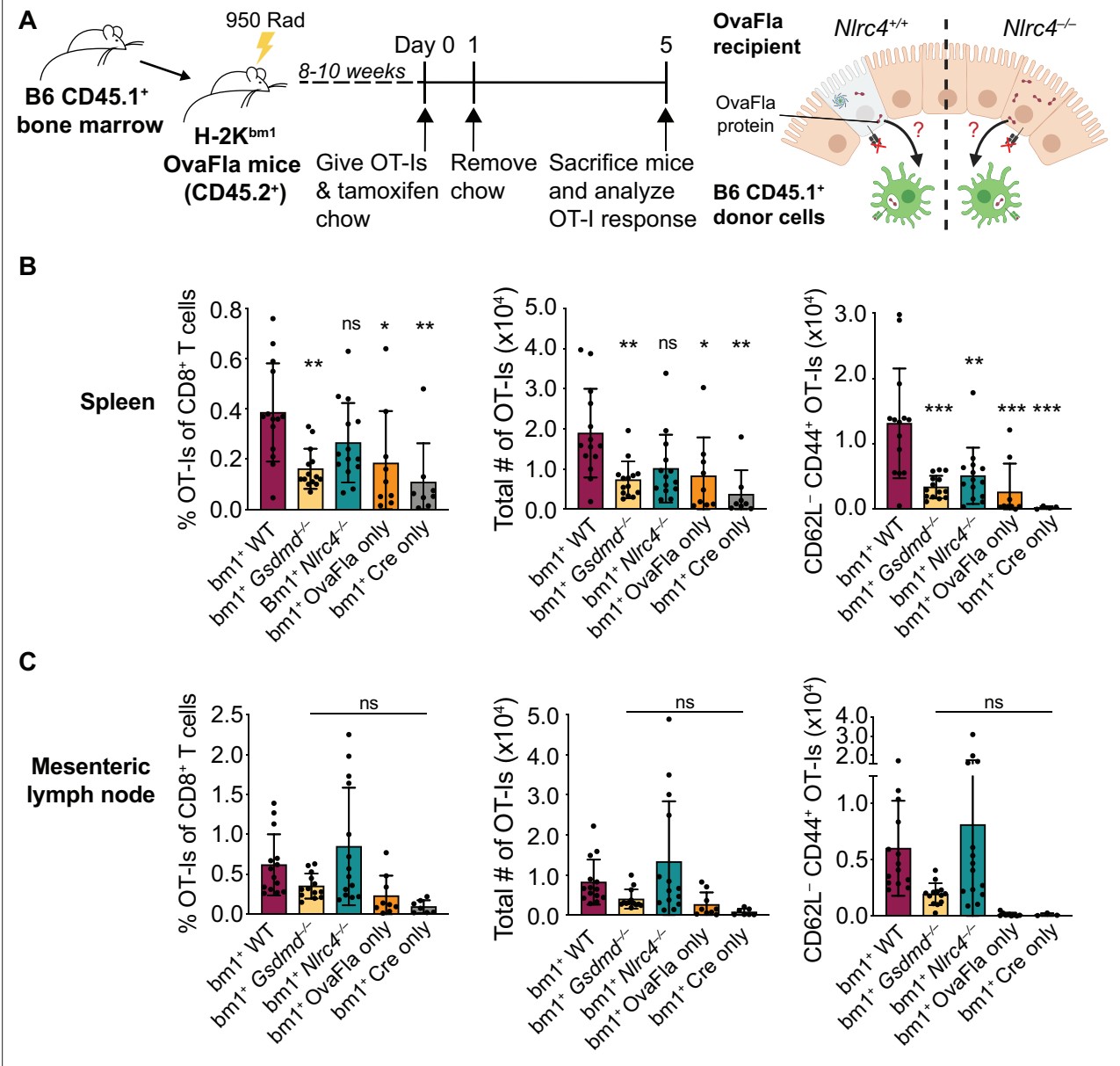

**Figure 4.** OvaFla expression in intestinal epithelial cells (IECs) results in OT-I cross-priming that is independent of NLRC4 but partially dependent on gasdermin D. (**A**) Schematic depicting the production and analysis workflow of chimeric bm1+OvaFla mice (left). At the right, an illustration of either wild-type (WT) OvaFla mice (left of the dashed line) or *Nlrc4*−/− OvaFla mice (right of the dashed line) following lethal irradiation and reconstitution with bone marrow from B6.SJL mice. (**B**) Quantification of OT-Is as a percent of total CD8+ T cells (left), the total number of OT-Is (middle), and the total number of CD62L−CD44+ OT-Is (right) in the spleen. (**C**) Quantification of OT-Is as a percent of total CD8+ T cells (left), the total number of OT-Is (middle), and the total number of CD62L−CD44+ OT-Is (right) in the mesenteric lymph nodes. Tissues were harvested and analyzed at day 5 post tamoxifen chow start. (**B–C**) Data are pooled from three biological replicates, and each dot represents an individual mouse. Data shown as mean ± SD. Significance calculated using one-way ANOVA and Tukey's multiple comparisons test (*$p < 0.05$, **$p < 0.01$, ***$p < 0.001$). Only p values between WT and other experimental groups are shown. See *Figure 4—source data 1* for exact p values.

The online version of this article includes the following source data and figure supplement(s) for figure 4:

**Source data 1.** Statistical data for *Figure 4*.

**Figure supplement 1.** OvaFla expression in intestinal epithelial cells (IECs) of bm1+OvaFla mice results in NAIP–NLRC4 expression and OT-I cross-priming that is independent of NLRC4 but partially dependent on gasdermin D.

**Figure supplement 1—source data 1.** Statistical data for *Figure 4—figure supplement 1*.

**Figure supplement 2.** K^b donor bone marrow is required for OT-I proliferation and activation in bm1+ OvaFla bone marrow chimeras.

As with the non-chimera experiments, an obvious dividing and activated OT-I population was observed by flow cytometry in each of the OvaFla mouse lines (*Figure 4B and C*, *Figure 4—figure supplement 1B, C*). This population was absent in mice given H-2K$^{bm1}$ bone marrow (*Figure 4—figure supplement 2*), confirming the requirement for APCs to express K$^b$ to activate OT-I T cells. These data provide formal genetic evidence that IEC-derived antigens can be cross-presented to activate CD8$^+$ T cells in vivo.

In the spleen, the bm1$^+$ WT and bm1$^+$ *Nlrc4$^{-/-}$* OvaFla mice harbored significantly more OT-I T cells than the bm1$^+$ *Gsdmd$^{-/-}$* OvaFla mice, or the bm1$^+$ OvaFla-only and bm1$^+$ Cre-only littermate controls, by both percent (*Figure 4B*, left) and total number (*Figure 4B*, middle). There were also significantly more activated (CD62L$^-$CD44$^+$) OT-I T cells in the spleens of bm1$^+$ WT mice as compared to bm1$^+$ *Gsdmd$^{-/-}$* mice (*Figure 4B*, right). In the mesenteric lymph nodes, no significant differences were found across any of the OvaFla mouse lines (*Figure 4C*, *Figure 4—figure supplement 1C-D*). The reason for the weak responses in the mesenteric lymph nodes is unclear, but others have previously noted negative impacts in irradiation chimeras on the expansion of adoptively transferred OT-I T cells (*Kurts et al., 1997*).

Taken together, these data provide genetic evidence that OT-I T cells are cross-primed from IEC-derived antigen following OvaFla induction. This cross-priming does not strictly require NAIP–NLRC4 activation, but gasdermin D-induced pyroptosis can promote CD8$^+$ T cell responses, at least for splenic OT-I T cells.

## NAIP–NLRC4 activation drives *Batf3$^+$* cDC1-independent cross-presentation

Previous work shows that ex vivo cDC1s can cross-prime CD8$^+$ T cells with IEC-derived antigen (*Cerovic et al., 2015*). To investigate the role of cDC1s in vivo, we first compared the relative number (*Figure 5—figure supplement 1B* left) and maturation state (MHC II$^{high}$ CD86$^+$) (*Figure 5—figure supplement 1B* right) of cDC1s in the mesenteric lymph nodes and spleen across the WT, *Pycard$^{-/-}$*, *Gsdmd$^{-/-}$*, and *Nlrc4$^{-/-}$* OvaFla mice after 2 days of tamoxifen chow. Although there was a modest reduction in the relative number of cDC1s in the spleens of *Pycard$^{-/-}$* OvaFla mice relative to the WT OvaFla mice, there was otherwise no clear difference in the presence or maturation state of cDC1s across the various OvaFla lines. These data suggest that NAIP–NLRC4 activation in IECs does not have a broad impact on cDC1s.

However, it is possible that a relatively small number of cDC1s are receiving antigen and maturation signals in our OvaFla model, so we assessed whether cDC1s are required for cross-priming OT-Is by genetically eliminating cDC1s. To do so, we used mice deficient for *Batf3*, a gene encoding a transcription factor required for development of XCR1$^+$ cross-presenting cDC1s (*Hildner et al., 2008*; *Lukowski et al., 2021*). We took advantage of our H-2K$^{bm1}$ bone marrow chimera system and compared bm1$^+$ OvaFla recipients that received either B6 CD45.1 bone marrow or bone marrow from *Batf3$^{-/-}$* mice.

As with the above experiments, bone marrow chimeras were made by lethally irradiating bm1$^+$ OvaFla mice and transferring donor bone marrow from either B6 CD45.1 or *Batf3$^{-/-}$* donors. Eight to ten weeks post irradiation, $2 \times 10^4$ CD45.1$^+$ CD45.2$^+$ CellTrace Violet labeled OT-I T cells were adoptively transferred intravenously, and the mice were given a 1-day pulse of tamoxifen chow (as in *Figure 4A*, left). The mice were sacrificed 5 days later, and their spleens and mesenteric lymph nodes were analyzed for OT-I proliferation and activation. We confirmed an absence of cDC1 cells in the OvaFla mice that received *Batf3$^{-/-}$* donor bone marrow (*Figure 5A*, *Figure 5—figure supplement 2*).

To our surprise, there was no difference in the relative (*Figure 5B–C*, top) or total (*Figure 5B–C*, middle) number of OT-I T cells between bm1$^+$ WT OvaFla mice that received B6 CD45.1 or *Batf3$^{-/-}$* bone marrow in either the spleen or mesenteric lymph node. OT-I T cells in these two mouse groups also appeared to proliferate similarly (*Figure 5D*, *Figure 5—figure supplement 3*). Additionally, there was no difference in the percent (*Figure 5—figure supplement 4*) or total number (*Figure 5B–C*, bottom) of CD44$^+$ CD62L$^-$ OT-I T cells. These data suggest a *Batf3*-independent population of DCs are responsible for cross-presentation of IEC-derived antigen following NAIP–NLRC4 activation.

The above findings with WT OvaFla mice are in stark contrast to the *Nlrc4$^{-/-}$* OvaFla mice, which exhibit a significant decrease in the relative (*Figure 5B–C*, top) and total (*Figure 5B–C*, middle) number of OT-I T cells in the spleens and mesenteric lymph nodes of mice that received *Batf3$^{-/-}$* donor cells

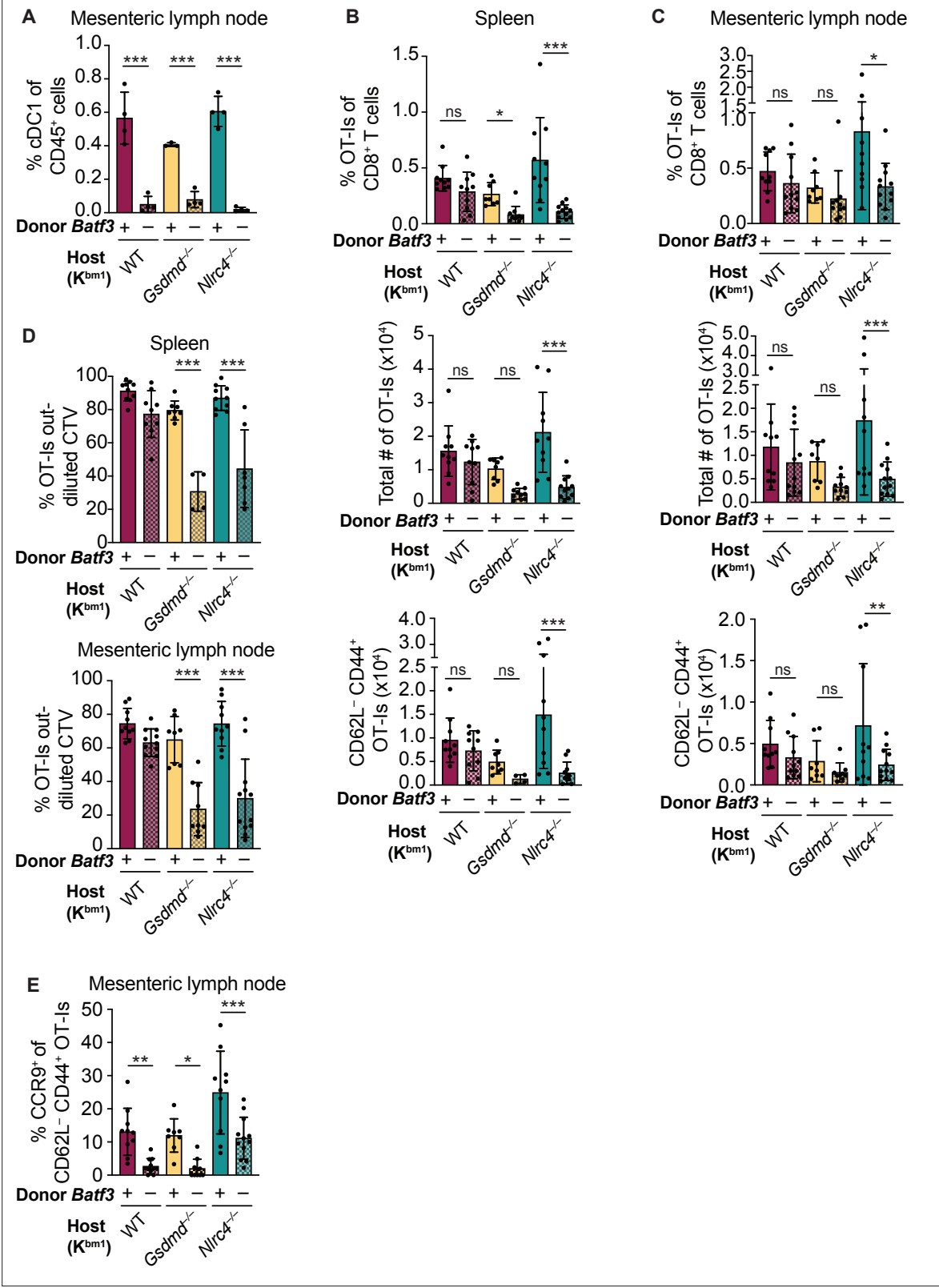

**Figure 5.** Cross-priming of OT-Is is independent of *Batf3*+ cDC1s following NAIP–NLRC4 activation in intestinal epithelial cells (IECs). (**A**) Percent of CD45+ cells that are conventional type one dendritic cells (cDC1s) in bm1 chimera mice that received either *Batf3*+ or *Batf3*− donor bone marrow. (**B**) Quantification of OT-Is as a percent of total CD8+ T cells (top), the total number of OT-Is (middle), and the total number of CD62L−CD44+ OT-Is (bottom) in the spleen. (**C**) Quantification of OT-Is as a percent of total CD8+ T cells (top), the total number of OT-Is (middle), and the total number of

*Figure 5 continued on next page*

*Figure 5 continued*

CD62L⁻CD44⁺ OT-Is (bottom) in the mesenteric lymph nodes. (**D**) Quantification of OT-Is that have out-diluted the CellTrace Violet dye in the spleen (top) and mesenteric lymph nodes (bottom). (**E**) Percent of CD62L⁻CD44⁺ OT-Is in the mesenteric lymph node that are CCR9⁺. Tissues were harvested and analyzed at day 5 post tamoxifen chow start. Samples with fewer than 20 OT-Is were excluded from CD62L, CD44, and CCR9 calculations. (A) Data are from a single experiment. (B–D) Data are pooled from three biological replicates. (E) Data are pooled from two biological replicates. Each dot represents an individual mouse. Data shown as mean ± SD. Significance calculated using one-way ANOVA and Šídák's multiple comparisons test (*p < 0.05, **p < 0.01, ***p < 0.001). See *Figure 5—source data 1* for exact p values.

The online version of this article includes the following source data and figure supplement(s) for figure 5:

**Source data 1.** Statistical data for *Figure 5*.

**Figure supplement 1.** NAIP–NLRC4 activation in intestinal epithelial cells (IECs) does not lead to an increase in relative numbers of cDCs or an increase in their maturation state in the mesenteric lymph nodes or spleen.

**Figure supplement 1—source data 1.** Statistical data for *Figure 5—figure supplement 1*.

**Figure supplement 2.** Gating demonstration for *Figure 5A*.

**Figure supplement 3.** Representative histograms demonstrating the dilution of CellTrace Violet dye of OT-Is from individual mice shown in *Figure 5D*.

**Figure supplement 4.** No difference in the percent of CD62L⁻CD444⁺ OT-I T cells or in the TNFα and IFNγ production between genotypes of bm1⁺ OvaFla mice.

**Figure supplement 4—source data 1.** Statistical data for *Figure 5—figure supplement 4*.

compared to the mice that received B6 CD45.1 donor cells. There was a corresponding significant decrease in the total number of CD44⁺ CD62L⁻ OT-I T cells (*Figure 5B–C*, bottom). The difference in OT-I numbers between these two groups of mice may be related to a relative decrease in proliferation of the OT-I T cells in the mice receiving *Batf3⁻/⁻* bone marrow, as evidenced by less dilution of the Cell-Trace Violet dye (*Figure 5D*, *Figure 5—figure supplement 3*). These data indicate that in the absence of NAIP–NLRC4 inflammasome activation, efficient cross-presentation of IEC-derived antigen in vivo requires XCR1⁺ cDC1s, but that this requirement is circumvented when the inflammasome is activated.

NAIP–NLRC4 activation might promote alternative (cDC1-independent) cross-presentation pathways by the pyroptotic release of antigen and/or inflammatory cytokines. To test whether gasdermin D is required for cDC1-independent cross-priming, we examined bm1⁺*Gsdmd⁻/⁻* chimeras reconstituted with *Batf3⁺* or *Batf3⁻/⁻* bone marrow. The bm1⁺ *Gsdmd⁻/⁻* OvaFla mice exhibit a phenotype that falls between the bm1⁺ WT and bm1⁺ *Nlrc4⁻/⁻* OvaFla mice, with the only significant differences between WT and *Batf3⁻/⁻* bone marrow recipients in the division of OT-I T cells (*Figure 5D*, *Figure 5—figure supplement 3*) and relative percent of OT-I T cells in the spleen (*Figure 5B*, top). These data suggest that the role for NAIP–NLRC4 activation in promoting *Batf3*-independent cross-presentation is minimally driven by IEC pyroptosis.

Regardless of the bone marrow donor, OT-I T cells in the bm1⁺ WT, bm1⁺ *Gsdmd⁻/⁻*, and bm1⁺ *Nlrc4⁻/⁻* OvaFla mice all showed similar levels of TNFα and IFNγ production following ex vivo stimulation with PMA and ionomycin (*Figure 5—figure supplement 4B*). However, when we looked at CCR9 expression as a readout of whether the OT-I T cells were homing to the intestine (*Svensson et al., 2002*), we found a significant decrease in the number of cells expressing CCR9 in the *Batf3⁻/⁻* recipients relative to the B6 CD45.1 recipients across all three mouse lines (*Figure 5E*). These data align with previous findings that show that cDC1s play a key role in driving CCR9 expression on CD8⁺ T cells (*Joeris et al., 2021*). In summary, our data indicate the existence of two potential pathways by which IEC-derived antigens are cross-presented to CD8⁺ T cells: a constitutive pathway that operates in the absence of inflammasome activation that requires *Batf3⁺* cDC1s, and a pathway that operates in the presence of inflammasome activation that does not require *Batf3⁺* cDC1s. Interestingly, the *Batf3⁺* cDC1s appear necessary for instructing antigen-specific CD8⁺ T cells back to the intestine.

## cDCs are required for cross-presentation of IEC-derived antigen

Although XCR1-expressing cDC1s are the dominant cross-presenting cell type (*Bachem et al., 2012*; *Dorner et al., 2009*), other APCs are reportedly capable of cross-priming CD8⁺ T cells as well. These APCs include monocyte-derived DCs (moDCs) (*Briseño et al., 2016*) and red pulp macrophages (*Enders et al., 2020*). Additionally, cDC2s have been show to acquire characteristics of cDC1s under inflammatory conditions (*Bosteels et al., 2020*) or in the absence of *Batf3* (*Lukowski et al., 2021*),

though it remains uncertain if these cells are able to cross-prime CD8+ T cells or provide T cells with the appropriate homing signals.

To determine whether our *Batf3*-independent cross-presenting population was another cDC population (e.g., cDC2s) or a macrophage or moDC population, we conducted a modified version of the above chimera experiments in which we compared bm1+ OvaFla mice that received bone marrow from either B6 CD45.1 mice or *Zbtb46*-DTR (diphtheria toxin [DT] receptor) mice (*Meredith et al., 2012*). *Zbtb46* is a transcription factor that drives development of cDCs but not moDCs, macrophages or any other myeloid cell populations (*Meredith et al., 2012*; *Satpathy et al., 2012*). Insertion of the DTR gene into the 3' untranslated region of *Zbtb46* allows for targeted ablation of these cells in bone marrow chimeras following DT treatment (*Meredith et al., 2012*). Eight weeks post bone marrow reconstitution, all mice were given DT 1 day prior to OT-I transfer and tamoxifen chow pulse and again 3 days later. As before, spleens and mesenteric lymph nodes were collected at day 5 post tamoxifen treatment and analyzed for evidence of cross-primed OT-I CD8+ T cells.

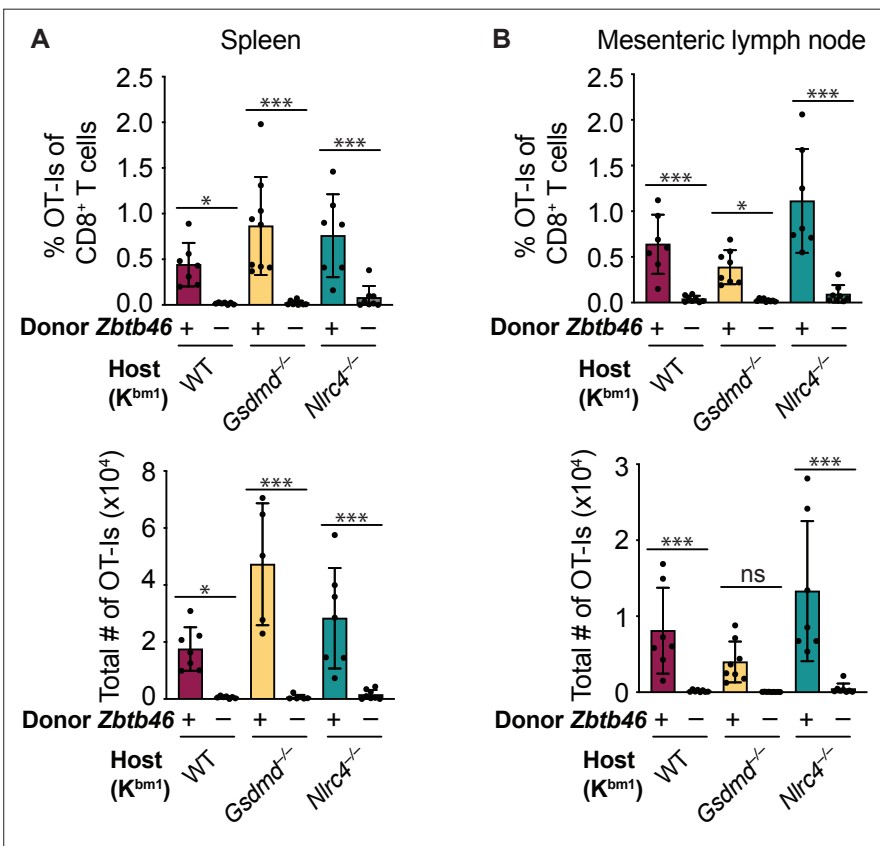

**Figure 6.** Cross-presentation of intestinal epithelial cell (IEC)-derived antigen is dependent on cDCs.
(**A**) Quantification of OT-Is as a percent of total CD8+ T cells (top) and the total number of OT-Is (bottom) in the spleen. (**B**) Quantification of OT-Is as a percent of total CD8+ T cells (top) and the total number of OT-Is (bottom) in the mesenteric lymph nodes. All mice were given two doses of diphtheria toxin (DT) (see Materials and methods), and tissues were harvested and analyzed at day 5 post tamoxifen chow start. Data are pooled from two biological replicates. Data shown as mean ± SD. Significance calculated using one-way ANOVA and Šídák's multiple comparisons test (*p < 0.05, **p < 0.01, ***p < 0.001). See *Figure 6—source data 1* for exact p values.

The online version of this article includes the following source data and figure supplement(s) for figure 6:

**Source data 1.** Statistical data for *Figure 5*.

**Figure supplement 1.** Gating strategy to identify conventional type one dendritic cells (cDC1s) and cDC2s in chimeric bm1+ OvaFla mice that received bone marrow from either B6 mice (top) or Zbtb46–DTR mice (bottom).

**Figure supplement 2.** Injection of diphtheria toxin (DT) results in the depletion of conventional type one dendritic cells (cDC1s) and cDC2s from chimeric bm1+ OvaFla mice given Zbtb46–DTR bone marrow.

**Figure supplement 2—source data 1.** Statistical data for *Figure 6—figure supplement 2*.

When we compared the *Zbtb46*[+] (B6) bone marrow recipients with the *Zbtb46*-DTR bone marrow recipients, we found a significant reduction of cDC1s and cDC2s in the mesenteric lymph nodes (*Figure 6—figure supplements 1 and 2*). Furthermore, both the relative (*Figure 6A–B*, top) and total (*Figure 6A–B*, bottom) numbers of OT-Is were significantly reduced in the mice that received *Zbtb46*-DTR bone marrow across all WT, *Gsdmd*[–/–], and *Nlrc4*[–/–] OvaFla mice. These data demonstrate that the *Batf3*-independent population of cross-presenting cells seen in the WT OvaFla mice in *Figure 5* are dependent on *Zbtb46* and thus are likely due to a non-cDC1 subset (presumably cDC2s).

## Discussion

IECs represent an important barrier surface that protects against enteric pathogens. At the same time, IECs also represent a potential replicative niche for pathogens. As such, the immune system must survey IECs for foreign antigens and present those antigens to activate protective adaptive immune responses. In general, it remains poorly understood whether and how IEC-derived antigens are presented to activate T cell responses. In particular, the relative contributions of direct versus cross-presentation of IEC antigens to CD8[+] T cells have not been thoroughly investigated. Here, we employed a genetic system that inducibly expresses a model antigen (ovalbumin) fused to an NAIP–NLRC4 agonist (flagellin) within the cytosol of cells (*Nichols et al., 2017b*). We crossed these 'OvaFla' mice to Villin-Cre-ER[T2] mice, allowing for tamoxifen-inducible expression specifically in IECs. By additionally crossing to an H-2K[bm1] background (*Nikolić-Žugić and Bevan, 1990*; *Schulze et al., 1983*), and using the resulting mice as irradiated recipients for WT K[b] hematopoietic donor cells, we engineered a system in which an IEC-derived ovalbumin antigen (SIINFEKL) cannot be directly presented to OT-I T cells but can still be acquired by hematopoietic cells and cross-presented. Using this system, we established in vivo that there is an antigen presentation pathway in which IEC-derived antigens are cross-presented to activate CD8[+] T cells. This finding extends previous work indicating that ex vivo isolated DCs can cross-present IEC-derived antigens to CD8[+] T cells (*Cerovic et al., 2015*; *Cummings et al., 2016*). We show that these antigens can activate antigen-specific CD8[+] T cells in vivo, and that this activation can occur even when direct presentation is genetically eliminated. We suggest that the cross-presentation pathway revealed by our analyses could be of importance during infection with pathogens that replicate in IECs, though future studies will be required to evaluate this hypothesis.

Our genetic system also allowed us to assess the contribution of IEC inflammasome activation to the adaptive immune response. Inflammasomes are a critical component of the innate immune response to many pathogens, and their activation is known to influence adaptive immunity (*Deets and Vance, 2021*). However, in previous studies, it has been difficult to isolate the specific effects of inflammasome activation on adaptive immunity because microbial pathogens activate numerous innate immune signaling pathways over the course of an infection. By providing a genetically encoded antigen and inflammasome stimulus, we were able to overcome this issue and specifically address the role of inflammasomes in adaptive CD8[+] T cell responses in vivo. We crossed our OvaFla Villin-Cre-ER[T2] mice to mice deficient in key inflammasome components. Consistent with previous work, we found that *Nlrc4*[–/–] mice entirely lack the inflammasome response to cytosolic flagellin, whereas *Pycard*[–/–] mice are defective for IL-18 release but not pyroptotic cell death or IEC expulsion (*Rauch et al., 2017*; *Figures 1C, E and 2B*). We also crossed OvaFla Villin-Cre-ER[T2] mice to pyroptosis-deficient *Gsdmd*[–/–] mice and found that they were defective for IL-18 release in vivo (*Figure 1C and E*).

Because *Nlrc4*[–/–] IECs fail to undergo pyroptosis or IEC expulsion (*Rauch et al., 2017*), we noted that cells expressing the OvaFla transgene accumulate to much higher levels in the *Nlrc4*[–/–] mice than in WT, *Pycard*[–/–], or *Gsdmd*[–/–] mice, in which IEC expulsion still occurs (*Figure 2*). Higher levels of Ova antigen in IECs has previously found to correlate with higher levels of OT-I expansion in the spleen and mesenteric lymph nodes of mice (*Vezys et al., 2000*). Because of the differences in antigen level, comparisons of *Nlrc4*[–/–] mice to the other genotypes must be made with caution. Nevertheless, we found that OT-I T cells in the *Nlrc4*[–/–] OvaFla mice divide and are activated at similar levels to the WT OvaFla mice following tamoxifen administration (*Figure 3B–D*). This activation occurred even when direct presentation of the OT-I peptide by IECs was eliminated on the K[bm1] background (*Figure 4B–C*). These results are surprising for two reasons. First, it is not clear how IEC-derived antigens would be delivered to APCs in the absence of inflammasome-induced cell death. Other studies have suggested that IEC apoptosis, which may occur during homeostatic IEC turnover (*Bullen et al., 2006*; *Hall et al., 1994*; *Marshman et al., 2001*; *Shibahara et al., 1995*; *Watson et al., 2005*), can be a source of

eLife Research article

antigen for T cell activation (*Cummings et al., 2016*; *Huang et al., 2000*). However, apoptotic IECs are expelled apically into the intestinal lumen (*Bullen et al., 2006*; *Hall et al., 1994*; *Marshman et al., 2001*; *Shibahara et al., 1995*; *Watson et al., 2005*), and so the exact mechanism of basolateral antigen delivery remains unclear—though it may involve luminal sampling by intestinal phagocytes (*Farache et al., 2013*) and/or the transfer of plasma membrane components (trogocytosis) (*Dance, 2019*). Cummings et al. suggested that IECs can be engulfed by APCs, resulting in antigen presentation on MHC class II to induce CD4$^+$ T regulatory cells, but this work did not examine antigen-specific responses or MHC class I presentation to CD8$^+$ T cells. Additionally, Joeris et al. recently showed that cDC1s can present IEC-derived self-antigen to drive cross-tolerant OT-I T cells (*Joeris et al., 2021*). Further work is therefore needed to understand mechanisms of IEC-derived antigen presentation in the absence of inflammatory cell death. The second reason we were surprised to see CD8$^+$ T cell activation in *Nlrc4*$^{-/-}$ OvaFla mice is that these mice are presumably unable to produce inflammatory signals necessary to induce APC activation. However, previous studies have shown that OT-I T cells can be activated from constitutively expressed Ova in the absence of inflammation. In this scenario, the CD8$^+$ T cells go on to become anergic and are likely eventually deleted from the periphery (*Kurts et al., 1996*; *Kurts et al., 1997*; *Liu et al., 2007*; *Vezys et al., 2000*).

Since WT, *Pycard*$^{-/-}$, and *Gsdmd*$^{-/-}$ IECs all undergo cell death and IEC expulsion in response to NLRC4 activation, these mice exhibit similar levels of OvaFla transgene expression in IECs, allowing for comparisons between these mouse strains (*Figure 2C*). We found that OvaFla production leads to CD8$^+$ T cell expansion and activation in all of these strains. The expansion is at least partially dependent on gasdermin D, as *Gsdmd*$^{-/-}$ OvaFla mice have significantly fewer OT-I T cells than their WT counterparts (*Figure 3B and C*). Interestingly, ASC-deficient OvaFla mice—in which IECs still undergo pyroptosis following NAIP–NLRC4 activation (*Rauch et al., 2016*)—show similar, or even increased, OT-I numbers in their tissues relative to WT OvaFla mice (*Figure 3B and C*). These data, combined with the fact that *Gsdmd*$^{-/-}$ and *Pycard*$^{-/-}$ OvaFla mice have little to no detectable IL-18 in their serum (*Figure 1C and E*), suggest that the difference in OT-I T cell proliferation between these strains is in some way related to pyroptotic antigen release. One hypothesis is that the gasdermin D pore, which has been shown to provide a lysis-independent portal for IL-1β, IL-18, and other small proteins (*DiPeso et al., 2017*; *Evavold et al., 2018*; *Heilig et al., 2018*), may act as a channel for small antigens to escape IECs prior to cell expulsion.

Because *Gsdmd* deficiency only modestly affected OT-I responses, our data additionally suggest that there may be both GSDMD-dependent and GSDMD-independent pathways by which IEC antigens can be cross-presented to CD8$^+$ T cells. Because *Batf3*$^{-/-}$-dependent cDC1s have a known role in cross-presenting IEC-derived antigen (*Cerovic et al., 2015*), we sought to determine if the cross-presentation occurring in the OvaFla mice similarly relied on these cells. We compared bm1$^+$ OvaFla mice that received B6 CD45.1 bone marrow with those that received bone marrow from *Batf3*-deficient mice (*Figure 5A*). To our surprise, we found OT-I T cells were cross-primed in the bm1$^+$ WT OvaFla mice, even in the recipients that lacked cDC1s (*Figure 5A–D*). Interestingly, these data contrast with the bm1$^+$ *Nlrc4*$^{-/-}$ OvaFla mice, where the recipients given *Batf3*-deficient bone marrow had significantly fewer activated OT-I T cells than their counterparts given *Batf3*-sufficient bone marrow. OT-I T cell activation in the bm1$^+$ *Gsdmd*$^{-/-}$ OvaFla mice partially relied on *Batf3*$^+$ DCs. Furthermore, Cell-Trace Violet data show the OT-I T cells in the bm1$^+$ *Nlrc4*$^{-/-}$ and bm1$^+$ *Gsdmd*$^{-/-}$ OvaFla mice undergo fewer rounds of division in the absence of *Batf3* cDC1s (*Figure 5D*). These data suggest there may be two possible cross-presentation pathways for IEC-derived antigen: one that occurs in the presence, and one in the absence, of inflammasome-derived inflammatory signals. We found that *Zbtb46*$^+$ bone marrow-derived cells were required for both pathways (*Figure 6*), indicating that cross-presentation seen under inflammatory conditions occurs through a *Batf3*-independent but *Zbtb46*-dependent cDC population. We hypothesize that these cells are cDC2s, as recent work shows that cDC2s can take on characteristics of cDC1s under inflammatory conditions (*Bosteels et al., 2020*) or in the absence of *Batf3* (*Lukowski et al., 2021*).

Our work raises several interesting questions for future study, including the mechanism of cDC maturation. The traditional model of DC maturation involves TLR signaling on the DC (*Dalod et al., 2014*). IL-1R (*Pang et al., 2013*) or IL-18R (*Li et al., 2004*) on these cells might also trigger maturation, though further investigation is needed to understand how IL-1β, IL-18, or other inflammatory signals,

such as eicosanoids (*McDougal et al., 2021*; *Oyesola et al., 2021*; *Rauch et al., 2017*), downstream of inflammasome activation might drive maturation of DCs that have acquired IEC-derived antigen.

Overall, our studies show that IEC-derived antigens are cross-presented both following NAIP–NLRC4 activation and under apparent homeostatic conditions in the absence of NAIP–NLRC4-induced inflammation. In the context of NAIP–NLRC4 activation, cross-priming of CD8+ T cells is partially dependent on gasdermin D-mediated pyroptosis and requires a *Batf3*-independent cDC population. These data add insights to the complex interactions between innate and adaptive immune responses occurring in the intestine.

# Materials and methods

**Key resources table**

| Reagent type (species) or resource | Designation | Source or reference | Identifiers | Additional information |
|---|---|---|---|---|
| Gene (*Mus musculus*) | *Nlrc4* | GenBank | Gene ID: 268973 | |
| Gene (*Mus musculus*) | *Gsdmd* | GenBank | Gene ID: 69146 | |
| Gene (*Mus musculus*) | *Pycard* | GenBank | Gene ID: 66824 | |
| Gene (*Mus musculus*) | *Batf3* | GenBank | Gene ID: 55509 | |
| Gene (*Mus musculus*) | *Zbtb46* | GenBank | Gene ID: 72147 | |
| Strain, strain background (*Mus musculus*) | *Nlrc4*−/− | PMID:15190255 | RRID:MGI:3047280 | Vishva Dixit, Genentech, South San Francisco, CA |
| Strain, strain background (*Mus musculus*) | *Gsdmd*−/− | PMID:28410991 | RRID:IMSR_JAX:032663 | Generated via CRISPR/Cas9 from UC Berkeley Gene Targeting Facility |
| Strain, strain background (*Mus musculus*) | *Pycard*−/− | PMID:15190255 | RRID:MGI:3047277 | Vishva Dixit, Genentech, South San Francisco, CA |
| Strain, strain background (*Mus musculus*) | *Batf3*−/− | Jackson Laboratory | RRID:IMSR_JAX:013755 | C57BL/6J background |
| Strain, strain background (*Mus musculus*) | *Zbtb46*−/− | Jackson Laboratory | RRID:IMSR_JAX:019506 | |
| Strain, strain background (*Mus musculus*) | Villin-Cre-ER^T2 | Jackson Laboratory | RRID:IMSR_JAX:020282 | C57BL/6NJ background |
| Strain, strain background (*Mus musculus*) | OT-I *Rag2*−/− | Jackson Laboratory | RRID:IMSR_JAX:003831 | C57BL/6 background |
| Genetic reagent (*Mus musculus*) | OvaFla | PMID:29263322 | MGI:6196853 | |
| Antibody | CD16/CD32 Purified (rat monoclonal) | eBioscience | Clone: 93; Cat#: 14-0161-85 | FC(1:1000) |
| Antibody | Anti-mouse CD45.1 APC (mouse monoclonal) | eBioscience | Clone: A20; Cat#: 17-0453-81 | FC(1:300) |
| Antibody | Anti-mouse CD45 APC (rat monoclonal) | Biolegend | Clone: 30-F11; Cat#: 103111 | FC(1:300) |
| Antibody | Anti-mouse CD45.2 PE/Cy7 (mouse monoclonal) | BioLegend | Clone: 104; Cat#: 109830 | FC(1:300) |
| Antibody | Anti-mouse CD8a Brilliant Violet 650 (rat monoclonal) | BioLegend | Clone: 53–6.7; Cat#: 100742 | FC(1:300) |
| Antibody | Anti-mouse CD44 BB515 (rat monoclonal) | BD | Clone: IM9; Cat#: 564587 | FC(1:300) |
| Antibody | Anti-mouse CD62L Brilliant Violet 711 (rat monoclonal) | BioLegend | Clone: MEL-14; Cat#: 104445 | FC(1:300) |
| Antibody | Anti-mouse CD199 (CCR9) PE (rat monoclonal) | BioLegend | Clone: 9B1; Cat#: 129707 | FC(1:100) |

*Continued on next page*

*Continued*

| Reagent type (species) or resource | Designation | Source or reference | Identifiers | Additional information |
|---|---|---|---|---|
| Antibody | Anti-mouse TNFa FITC (mouse monoclonal) | eBioscience | Clone: MP6-XT22; Cat#: 11-7321-82 | FC(1:100) |
| Antibody | Anti-mouse CD11c PE (arm hamster monoclonal) | eBioscience | Clone: 418; Cat#: 12-0114-81 | FC(1:300) |
| Antibody | Anti-mouse MHC Class II (I-A/I-E) APC-eFluor 780 (rat monoclonal) | BioLegend | Clone: M5/114.15.2; Cat#: 107628 | FC(1:300) |
| Antibody | Anti-mouse CD4 APC/Fire 750 (rat monoclonal) | BioLegend | Clone: GK1.5; Cat#: 100460 | FC(1:300) |
| Antibody | Anti-Mouse CD11b PE-Cyanine7 (rat monoclonal) | eBioscience | Clone: M1/70; Cat#: 25-0112-81 | FC(1:300) |
| Antibody | Anti-mouse CD11c Brilliant Violet 711 (arm hamster monoclonal) | BioLegend | Clone: N418; Cat#: 117349 | FC(1:300) |
| Antibody | Anti-mouse CD45 Brilliant Violet 785(rat monoclonal) | BioLegend | Clone: 30-F11; Cat#: 103149 | FC(1:300) |
| Antibody | Anti-mouse MHC II I-A/I-E FITC (rat monoclonal) | BioLegend | Clone: M5/114.15.2; Cat#: 107605 | FC(1:400) |
| Antibody | Anti-mouse/rat XCR1 APC (mouse monoclonal) | BioLegend | Clone: ZET; Cat#: 148206 | FC(1:300) |
| Antibody | Anti-mouse CD90.2 (Thy-1.2) APC/Fire 750 (rat monoclonal) | BioLegend | Clone: 53–2.1; Cat#: 140326 | FC(1:300) |
| Antibody | Anti-mouse Ly-6G/Ly-6C (Gr-1) APC/Cyanine7 (rat monoclonal) | BioLegend | Clone: Gr1; Cat#: 108424 | FC(1:300) |
| Antibody | Anti-mouse CD64 (FcγRI) APC (mouse monoclonal) | BioLegend | Clone: X54-5/7.1; Cat#: 139306 | FC(1:100) |
| Antibody | Anti-mouse CD45.2 PerCP-Cyanine5.5 (mouse monoclonal) | eBio | Clone: 45-0454-82; Cat#: 17-0454-82 | FC(1:100) |
| Antibody | Anti-mouse MHC Class II (I-A/I-E) FITC (rat monoclonal) | Fisher | Clone: M5/114.15.2; Cat#: 11-5321-82 | FC(1:300) |
| Antibody | Anti-mouse CD64 PE (mouse monoclonal) | Fisher | Clone: X54-5/7.1; Cat#: 12-0641-82 | FC(1:200) |
| Antibody | Anti-mouse CD45.2 PE (mouse monoclonal) | Fisher | Clonne: 104; Cat#: 12-0454-82 | FC(1:300) |
| Antibody | Anti-mouse CD11b PE-Cyanine7 (rat monoclonal) | Fisher | Clone: M1/70; Cat#: 25-0112-82 | FC(1:300) |
| Antibody | Anti-mouse CD90.2 (Thy-1.2) Pacific Blue (rat monoclonal) | BioLegend | Clond: 53–2.1; Cat#: 140306 | FC(1:300) |
| Antibody | Anti-mouse CD86 Brilliant Violet 785 (rat monoclonal) | BioLegend | Clone: GL-1; Cat#: 105043 | FC(1:200) |
| Antibody | Anti-mouse CD172a (SIRPα) Brilliant Violet 510 (rat monoclonal) | BioLegend | Clone: P84; Cat#: 144032 | FC(1:200) |
| Antibody | Ghost Dye Red 780 | Tonbo | Cat#: 13–0865T500 | FC(1:1000) |
| Antibody | Anti-rabbit IgG (H + L) AF 488 (donkey polyclonal) | Jackson Immunoresearch | Cat#: 711-545-152 | IF(1:500) |
| Antibody | Anti-mouse GFP Polyclonal Antibody (rabbit polyclonal) | Invitrogen | Cat#: A-6455 | IF(1:300) |
| Antibody | Anti-mouse IL-18 Biotin (rat monoclonal) | MBL | Clone: 93–10 C; Cat#: D048-6 | ELISA(1 μg/mL) |

*Continued on next page*

*Continued*

| Reagent type (species) or resource | Designation | Source or reference | Identifiers | Additional information |
|---|---|---|---|---|
| Antibody | Anti-mouse IL-18 (rat monoclonal) | BioXcell | Clone: YIGIF74-1G7; Cat#: BE0237 | ELISA(1:2000) |
| Antibody | Anti-mouse CD3 biotin (arm ham monoclonal) | BioLegend | Clone: 145–2 C11; Cat#: 100304 | For depletion, 10 µL/$10^7$ cells |
| Antibody | BD Pharmingen Streptavidin HRP | BD Biosciences | RRID:AB_2868972; Cat#: 554066 | ELISA(1:1000) |
| Commercial assay or kit | CellTrace Violet Cell Proliferation Kit | ThermoFisher | Cat#: C34557 | See Materials and methods section; 1 µL/$10^6$ cells |
| Commercial assay or kit | Anti-Biotin MicroBeads | Miltenyi | Cat#: 130-105-637 | For depletion, 20 µL/$10^7$ cells |
| Commercial assay or kit | LD Columns | Miltenyi | Cat#: 130-042-901 | See Materials and methods section |
| Chemical compound, drug | DAPI | Invitrogen | Cat#: D1306 | IF(10 nM) |
| Chemical compound, drug | BD GolgiPlug | BD Biosciences | Cat#: 555029 | FC(1:1000) |
| Chemical compound, drug | Phorbol myristate acetate (PMA) | Invivogen | Cat#: tlrl-pma | FC(1 µg/mL) |
| Chemical compound, drug | Ionomycin | Calbiochem | Cat#: 407952–1MG | FC(1 µg/mL) |
| Chemical compound, drug | o-Phenylenediamine dihydrochloride | Sigma | Cat#: P3804-100TAB | ELISA(one tab/ 5 mL) |
| Chemical compound, drug | Tamoxifen chow | envigo | Cat#: 130856 | See Materials and methods section |
| Chemical compound, drug | Diphtheria toxin from *Corynebacterium diphtheriae* | Sigma | Cat#: D0564-1MG | See Materials and methods section |
| Software, algorithm | ImageJ | NIH | RRID:SCR_003070 | |
| Software, algorithm | FlowJo | BD | RRID:SCR_008520 | |
| Software, algorithm | Prism | GraphPad | RRID:SCR_002798 | |

## Animals

All mice were maintained under specific pathogen-free conditions and, unless otherwise indicated, fed a standard chow diet (Harlan irradiated laboratory animal diet) ad libitum. OvaFla mice were generated as previously described (*Nichols et al., 2017b*) and crossed to Villin-Cre-ER^T2, which we obtained from Avril Ma (UCSF, San Francisco, CA) (*el Marjou et al., 2004*). OvaFla Villin-Cre-ER^T2 mice were additionally bred to *Gsdmd*^−/−, *Pycard*^−/−, and *Nlrc4*^−/− mice. *Nlrc4*^−/− and *Pycard*^−/− mice were from V. Dixit (*Mariathasan et al., 2004*) (Genentech, South San Francisco, CA). *Gsdmd*^−/− mice were previously described (*Rauch et al., 2017*). OT-I *Rag2*^−/− mice (from E Robey, Berkeley, CA) were used as a source of OT-Is for all adoptive transfer experiments.

For chimera experiments, the above OvaFla lines were crossed to B6.C-*H-2K*^bm1/ByJ mice (*Schulze et al., 1983*) (Jax strain 001060). For the bone marrow donors, B6.CD45.1 (Jax strain 002014), *Batf3*^−/− (Jax strain 013755), and *Zbtb46*–DTR (Jax strain 019506) mice were used.

Mice used for non-chimera experiments were 8–12 weeks of age upon tissue harvest, and mice used as chimeras were 16–20 of weeks of age upon tissue harvest. Female mice were co-housed, and all experimental mice were age- and sex-matched when possible. OvaFla-only and Cre-only controls were littermates of the experimental mice. All animal experiments and endpoints were approved by and performed in accordance with the regulations of the University of California Berkeley Institutional Animal Care and Use Committee.

## Adoptive transfer of OT-I T cells

The spleen and mesenteric lymph nodes were harvested from OT-I $Rag2^{-/-}$ mice, mashed between the frosted ends of two glass slides to create a single cell suspension, filtered through 100 µm nylon mesh, and pooled into a single tube. Red blood cells were lysed with ACK Lysing Buffer (Gibco; A10492-01). Cells were labeled with CellTrace Violet (ThermoFisher; C34557) following the manufacturer's protocol and transferred i.v. to mice anesthetized with isoflurane at $2 \times 10^4$ cells per mouse.

## Tamoxifen administration

The tamoxifen chow used in these studies was purchased from Envigo (https://www.envigo.com/tamoxifen-custom-diets; 120856). The diet contains 250 mg of tamoxifen per kilogram of chow and was irradiated prior to shipping. Mice were fed ab libitum for 2 days for the experiments in *Figure 1B–C*, *Figure 5—figure supplement 1* and 1 day for the remaining experiments. Envigo assumes approximately 40 mg of tamoxifen is consumed per kilogram of body weight per day for each mouse, though feed aversion leads to variable and limited initial food intake (*Chiang et al., 2010*).

## DT treatment

To deplete cDCs in the *Zbtb46*–DTR → bm1$^+$OvaFla chimeras, all mice were given two doses of DT (Sigma; D-0564) as described in *Meredith et al., 2012*. Each animal was given an initial dose of 20 ng DT per gram body weight 1 day prior to OT-I T cell transfer and tamoxifen chow pulse. The mice were then given a second dose of 4 ng DT per gram body weight 3 days after the initial dose.

## Flow cytometry

Spleens and mesenteric lymph nodes were harvested from euthanized mice and stored on ice in T cell media: RMPI 1640 (Gibco; 21870092) containing 10% FBS (Gibco, Cat#16140071, Lot#1447825), 1% penicillin-streptomycin, 1% L-glutamine, 1% sodium pyruvate, 0.5% 2-mercaptoethanol, and 25 mM HEPES. For lymphocyte staining, tissues were mashed between the frosted ends of glass slides and filtered through 100 µm nylon mesh. For myeloid staining, tissues were minced with scissors and forceps and incubated in T cell media containing 1 mg/mL collagenase VIII (Sigma; C2139-1G) or in HBSS (Ca$^{2+}$, Mg$^{2+}$) (Gibco; 14025076) containing DNase I (900 mg/1 mL) (Sigma; DN25-10MG) and Liberase TM (Roche; 5401119001), at 37°C for 25–45 min. The digested tissues were then passed through 70 µm filters and washed with T cell media. For all stains, red blood cells were lysed from a single cell suspension using ACK Lysing Buffer. Cells were counted using a Beckman Vi-CELL XR Cell Viability Analyzer (Beckman Coulter, Brea, CA), and $3 \times 10^6$ cells per tissue per mouse were added to individual FACS tubes or wells of a 96-well non-tissue culture-treated round bottom plate.

For extracellular surface staining, cells were blocked for 20–30 min with a 1:1000 dilution of anti-mouse CD16 and CD32 antibodies (eBioscience; 14-0161-85) at 4°C and then stained with a cocktail of antibodies for extracellular markers (Key resource table) at room temperature (RT) for 1 hr. All dilutions and washes were done with 1× PBS (Gibco; 10010049) containing 5% FBS/FCS.

For intracellular cytokine analysis, cells were incubated at $1 \times 10^6$ cells/mL T cell media plus 1 µg/mL PMA (Invivogen; tlrl-pma), 1 µg/mL ionomycin (Calbiochem; 407952–1 MG), and 1 µg/mL GolgiPlug (BD Biosciences; 555029) at 37°C for 5 hr. Cells were then washed and blocked for 20–30 min with a 1:1000 dilution of anti-mouse CD16 and CD32 antibodies at 4°C, and a surface stain was applied for 1 hr at RT (see Key resources table). Cells were then fixed in 100 mL eBioscience IC Fixation Buffer (Thermo; 00-8222-49) for 20–60 min RT, and then stained with an intracellular staining cocktail (see Key resources table) in 1× eBioscience Permeabilization Buffer (Thermo; 00-8333-56) at RT for 1 hr. Cells were washed and resuspended in PBS prior to analysis. The data were collected on a BD Biosciences Fortessa (San Jose, CA) in the UC Berkeley Cancer Research Laboratory Flow Cytometry facility, and analysis was performed using FlowJo 10 Software (BD Biosciences, San Jose, CA).

## Generation of bone marrow chimeras

Eight- to twelve-week-old mice were lethally irradiated with a Precision X-Rad320 X-ray irradiator (North Branford, CT) using a split dose of 500 rads and then 450 rads, approximately 15 hr apart. Bone marrow was harvested from the long bones of the indicated donor strains, red blood cells were lysed using ACK Lysing Buffer, and CD3$^+$ cells were depleted from the donor cells using a biotinylated anti-mouse CD3ε mAb (BioLegend; 100304) and the Miltenyi MACS MicroBead (Miltenyi;

130-105-637) magnetic depletion protocol with LD columns (Miltenyi; 130-042-901) to reduce graft versus host reactions (*Selvaggi et al., 1996*). Recipient mice were anesthetized with isoflurane, and approximately $5 \times 10^6$ donor cells were injected retro-orbitally. Females from the different strains were co-housed, and at least 8 weeks passed between reconstitution and the start of any experiment.

## Immunofluorescence

Mice were fed a single day pulse of tamoxifen chow and euthanized 2 days from start of the chow feeding. Approximately 2.5 cm pieces were taken from the proximal and distal ends of the small intestine. These pieces were flushed and fixed in PLP buffer (0.05 M phosphate buffer containing 0.1 M L-lysine [pH 7.4], 2 mg/mL NaIO$_4$, and 1% PFA) overnight at 4°C. The following day, tissues were washed 2× in phosphate buffer and placed in 30% sucrose overnight at 4°C. Tissue was frozen in Tissue-Tek OCT (VWR; 25608–930), cut on a Leica cryostat, and sections were placed on Fisherbrand Tissue Path Superfrost Plus Gold Slides (Fisher Scientific; 15-188-48).

For staining, slides were allowed to warm to RT, traced with an ImmEdge Hydrophobic Barrier Pen (Vector Labs; H-4000), washed 3× in 1× PBS with 0.5% Tween-20, and blocked with 10% normal donkey serum (Sigma; D9663) in 0.5% Tween-20, 100 mM TrisHCl [pH 7.5], 150 mM NaCl, 0.5% blocking reagent (Perkin Elmer; FP1020) for 30 min. Tissues were then stained with 1:300 GFP poly-clonal antibody (Invitrogen; A-6455) overnight at 4°C. Slides were washed 3× and stained with donkey anti-rabbit Alexa Fluor 488 (Jackson Immunoresearch; 711-545-152) for 60 min at RT, followed by 150 nM Acti-stain 555 phalloidin (Cytoskeleton, Inc; PHDH1-A) and 100 mM DAPI (D1306) for 30 min at RT. Slides were then washed 2× in H$_2$O and sealed under glass coverslips prior to imaging. All antibody dilutions were done in 100 mM TrisHCl [pH 7.5], 150 mM NaCl, 0.5% blocking reagent; all washes were done in 1× PBS with 0.5% Tween-20.

Slides were imaged on a Zeiss LSM710 at the CNR Biological Imaging Facility at the University of California, Berkeley. Images were blinded and manually quantified for GFP+ IECs. For quantification of GFP+ cells, DAPI+ IECs were counted in at least 15 villi per mouse—DAPI+ cells were counted prior to revealing the GFP+ cells in the 488 channel. For quantification of amount GFP levels per IEC, ImageJ (National Institutes of Health) was used to trace and measure the mean pixel intensity in the GFP channel for individual GFP+ cells, with 12–20 cells per image. ImageJ was used to visualize images and globally adjust contrast and brightness for print quality following quantification.

## Serum IL-18 measurement

Thermo Scientific Immuno MaxiSorp ELISA plates (Thermo Fisher; 439454) were coated with 1 µg/mL anti-mouse IL-18 mAb (MBL; D048-6) overnight at 4°C, and blocked with 1× PBS containing 1% BSA for 2–4 hr at RT. Serum was diluted 1:5 in PBS with 1% BSA, added to the plate with a purified IL-18 standard, and incubated overnight at 4°C. A biotinylated anti-mouse IL-18 sandwich mAb (BioXcell; BE0237) was added at 1:2000 in PBS with 1% BSA and incubated for 1–2 hr at RT. BD Pharmingen Streptavidin HRP (BD Biosciences; 554066) was added at 1:1000 in PBS with 1% BSA. Following a final 5× wash, plates were developed with 1 mg/mL OPD (Sigma; P3804-100TAB) in citrate buffer (PBS with 0.05 M NaH$_2$PO$_4$ and 0.02 M citric acid) plus 9.8 M H$_2$O$_2$. The reaction was stopped with a 3 M HCl acid stop after approximately 10 min. Absorbance at 490 nm was measured on a Tecan Spark multimode microplate reader (Tecan Trading AG, Switzerland).

## Statistical analysis

For all bar graphs, data are shown as mean ± SD. For *Figures 1–4*, the significance between each genotype was calculated using one-way ANOVA and Tukey's multiple comparisons test. For *Figure 5*, the significance between mice that received B6 CD45.1 and mice that received *Batf3*$^{-/-}$ bone marrow was calculated using one-way ANOVA and Šídák's multiple comparisons test. *Figure 6*, the significance between mice that received B6 CD45.1 bone marrow and mice that received *Zbtb46*-DTR bone marrow was calculated using one-way ANOVA and Šídák's multiple comparisons test. For all data, *p < 0.05, **p < 0.01, ***p < 0.001. Tests were run using GraphPad Prism (San Diego, CA).

The sample size for each experiment ranged from three to five mice per genotype, and two to three biological replicates (independent experiments) were performed per experiment, as indicated in figure legends. Sample size was chosen to provide the highest number of data points within the technical limitations of the tissue processing during the experiment.

## Acknowledgements

We thank members of the Vance and Barton Labs for discussions, Greg Barton and Ellen Robey for comments on the manuscript, Dmitri Kotov for comments on the manuscript and advice, and Roberto Chavez and Joceline Morales for technical assistance. We thank Jakob von Moltke for initial generation of the OvaFla mice. We also thank the UC Berkeley Cancer Research Laboratory Flow Cytometry facility, including Hector Nolla and Alma Valeros. *Figure 4A* was created with BioRender.com. Research reported in *Figure 2* of this publication was supported in part by the National Institutes of Health S10 program under award number 1S10RR026866-01. The content is solely the responsibility of the authors and does not necessarily represent the official views of the National Institutes of Health. REV is an Investigator of the Howard Hughes Medical Institute, and research in his laboratory is funded by NIH grants AI075039, AI063302, and AI155634.

## Additional information

### Competing interests

Russell E Vance: consults for Ventus Therapeutics and Tempest Therapeutics and is a Reviewing Editor for eLife. The other authors declare that no competing interests exist.

### Funding

| Funder | Grant reference number | Author |
| --- | --- | --- |
| National Institutes of Health | AI075039 | Russell E Vance |
| National Institutes of Health | AI063302 | Russell E Vance |
| National Institutes of Health | AI155634 | Russell E Vance |
| Howard Hughes Medical Institute | Investigator Award | Russell E Vance |
| National Institutes of Health | 5T32GM007232 | Katherine A Deets |

The funders had no role in study design, data collection and interpretation, or the decision to submit the work for publication.

### Author contributions

Katherine A Deets, Conceptualization, Formal analysis, Investigation, Methodology, Validation, Visualization, Writing – original draft, Writing – review and editing; Randilea Nichols Doyle, Conceptualization; Isabella Rauch, Conceptualization, Writing – review and editing; Russell E Vance, Conceptualization, Funding acquisition, Resources, Supervision, Writing – original draft, Writing – review and editing

### Author ORCIDs

Katherine A Deets (iD) http://orcid.org/0000-0003-1731-1641
Russell E Vance (iD) http://orcid.org/0000-0002-6686-3912

### Ethics

This study was performed in strict accordance with the recommendations in the Guide for the Care and Use of Laboratory Animals of the National Institutes of Health. Animal studies were approved by the UC Berkeley Animal Care and Use Committee (current protocol number: AUP-2014-09-6665-2).

### Decision letter and Author response

Decision letter https://doi.org/10.7554/eLife.72082.sa1
Author response https://doi.org/10.7554/eLife.72082.sa2

## Additional files

### Supplementary files
• Transparent reporting form

### Data availability
Immunofluorescence images have been deposited in Dryad and can be found at https://doi.org/10.6078/D1ST46. All remaining data generated or analyzed during this study are included in the manuscript and supporting files; Source Data files have been provided for Figures 1-6, Figure 3-figure supplement 1, Figure 4-figure supplement 1, Figure 5-figure supplement 1, Figure 5-figure supplement 4, Figure 6-figure supplement 2.

The following dataset was generated:

| Author(s) | Year | Dataset title | Dataset URL | Database and Identifier |
|---|---|---|---|---|
| Deets KA, Vance RE | 2021 | Data from: Inflammasome activation leads to cDC1-independent cross-priming of CD8 T cells by epithelial cell derived antigen | https://dx.doi.org/10.6078/D1ST46 | Dryad Digital Repository, 10.6078/D1ST46 |

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
