## [Editor Report]

This study uses sophisticated genetic tools to demonstrate that intestinal epithelial-derived antigens can be cross-presented by dendritic cells to activate CD8^+^ T cells via pyroptosis-dependent and -independent pathways. The study provides novel insight into how inflammasome activation regulates CD8 T cell responses. This paper will be of interest to scientists that are interested in both innate and adaptive immune systems.

---

## [Decision Letter]

**Decision letter after peer review:**

Thank you for submitting your article "Pyroptosis-dependent and -independent cross-priming of CD8^+^ T cells by intestinal epithelial cell-derived antigen" for consideration by *eLife*. Your article has been reviewed by 3 peer reviewers, including Chyung-Ru Wang as Reviewing Editor and Reviewer #1, and the evaluation has been overseen by Tadatsugu Taniguchi as the Senior Editor.

Essential revisions:

1. If would be of interest to see the effect of Asc, Gsdmd, Nlrc4-deficieny on CD4^+^ T cell responses in the same genetic model systems (e.g. co-transfer of OVA-specific CD4^+^ and CD8^+^ T cells into the same recipient).

2. The authors should comment on whether endogenous OVA-specific CD8^+^ T cell responses can be detected in these models (e.g. using tetramers or ELISPOT assays).

3. The differential expression of OVA, assessed by the GFP reporter in Figure 2, is concerning. The authors have a 2-variable system here, with mice contain different amounts of antigen and different pyroptosis-inducing potential. The authors acknowledge this point on line 216, but never do additional work to accurately quantify GFP expression across the genotypes of mice they are focused on. Additional quantification of transgene expression is critical, as the phenotypes described later in the study are modest.

4. Monitoring T cell response at day 5 post-treatment does not allow the authors to assess CD8 T cell differentiation into memory cells or functions in the context of infectious challenges with pathogens that express the OVA transgene. A suitable revision would include such analysis, where the authors examine later time points for T cell functions (20-40 days, for example) and the authors challenge these mice with pathogens (*Listeria*-OVA, for example), to assess functional CTL responses in a relevant setting.

5. While the OVA Fla+Villin ER-Cre model is elegant, it would have been better if the authors had also created a model to directly compare the steady state cross-presentation to CD8 T cells. The NLRC4 KO data suggest that there is steady state cross presentation happening in the absence of NLRC4 activation and there is no increase in cross-presentation because of NLRC4 activation and subsequent pyroptosis. Is it possible that the quality of primed CD8 T cells is different in the presence and absence of NLRC4. While the author examined IFNγ TNF production, granzyme B, Perforin expression or pathogen clearance experiments (*Listeria* OVA for example) would have tested the quality of primed CD8 T cells.

6. OT-I cross priming being independent of NLRC4 but partially dependent on gasdermin D is very interesting. Did the authors examine the MHC Class I-SIINFEKL complexes on DCs in the mesenteric lymph nodes by using either a clonotypic antibody (clone 25-D1.16) or in vitro priming of OT-I T cells by isolating DCs from mLNs. Similarly, was the activation status of cDC1s different between WT, Gasdermin DKO and NLRC4 KO mice after tamoxifen administration?

7. The study does point out that there is a Batf3 independent pathway of antigen cross -presentation to T cells that might be occurring upon activation of NLRC4 inflammasome however, the study overall might benefit from a better explanation of what other pathways or other populations of DCs might be facilitating the antigen presentation in this case. The authors do mention the recent study by Bosteel et al., briefly in the discussion but characterization of the inflammatory cDC2 population in the Nlrc4-/- might provide a better insight into the mechanism of antigen presentation.

8. While the study sets out to investigate the role of inflammasome activation and pyroptosis in CD8 T cell cross priming, based on the results, a more appropriate title would be" Inflammasome activation leads to cDC1 independent cross-priming of CD8 T cells by epithelial cell derived antigen".*Reviewer #1:*

In this study, the authors used Ova Fla Villin-Cre-ERT2 mouse model on various genetic backgrounds to demonstrate that IEC-derived antigens can be cross-presented by DC to activate CD8^+^ T cells via pyroptosis-dependent and -independent pathways. They also showed Batf3+ dendritic cells are uniquely required for the pyroptosis-independent antigen cross-presentation pathway. Overall, the experimental designs are sound and the conclusions are largely supported by the presented data.

Major strengths

– The experiments are well executed, and the results support their conclusions.

– The findings provide novel insight into how inflammasome activation affects CD8^+^ T cell responses.

– The design of genetic model systems used in this study is very logical.

– The data is clearly presented and well discussed.

Weaknesses:

– It is not clear whether the findings from OT-1 TCR Tg model can be extrapolated to polyclonal CD8^+^ T cells.

– This study would be more comprehensive if the authors could also investigate CD4^+^ T cell responses in the same genetic model systems.

– The authors should comment on some differential effects observed in the spleen and mesenteric lymph nodes.

*Reviewer #2:*

This study was focused on understanding the link between inflammasomes in intestinal epithelial cells and signals that stimulate adaptive immunity.

*Reviewer #3:*

This manuscript by Deets, K and co-authors describes the use of intricate in vivo genetic tools to understand the influence of NAIP-NLRC4 inflammasome activation and pyroptosis in IECs and subsequent activation of the adaptive immune system via cross-priming of CD8 T cells. With specific focus on the NAIP-NLRC4 inflammasome activation in IECs the authors use genetic tools leading to IEC specific expression of Ova-Fla that provides a cell specific expression of an antigen and an inflammasome activator to look at the subsequent effects on CD8 T cell priming in WT vs Asc-/-, Gsdmd-/- and Nlrc4-/- background. The authors find that NLRC4 inflammasome activation in IECs in fact is not necessary for OT-I T cell priming. The most interesting part of the manuscript are experiments described in figure 5 where BATF3 expressing cDC1s are critical for cross-priming in the absence of inflammasome activation but NLRC4 inflammasome activation bypasses the need for conventional cross-priming BATF3 expressing DCs. Finally, even though cross-priming of OT-I T cells may not require NLRC4 activation, there seems to be a minor role for Gasdermin D (and thus pyroptosis) that enhances OT-I T cell priming and expansion. The work is overall elegant and uses several sophisticated genetic tools to answer the question of how and whether inflammasome activation (in the absence of engagement of other PRRs such as TLRs) is critical for activation of adaptive immunity, specifically CD8 T cell priming here.

The weaknesses of the study are:

– The physiological relevance of the findings are unclear, since the authors fail to conduct any functional experiments, vis a vis quality of primed CD8 T cells in different scenarios (GSDMD KO, NLRC4 KO) to assess pathogen clearance.

– Failure to examine the density of peptide MHC complexes or the ability of different DCs to present antigen in vitro as a proxy for peptide MHC complexes.

– Lack of any insights into which DCs are cross-priming in the WT mice when BATF3 DCs are absent.

---

## [Author Response]

Essential revisions:1. If would be of interest to see the effect of Asc, Gsdmd, Nlrc4-deficieny on CD4^+^ T cell responses in the same genetic model systems (e.g. co-transfer of OVA-specific CD4^+^ and CD8^+^ T cells into the same recipient).

We agree that it would be very interesting to understand and compare the responses between OVA-specific CD8^+^ and CD4^+^ T cells. To address this question, we adoptively transferred OVA-specific CD4^+^ T cells (OT-IIs) into WT, *Gsdmd^–/–^*, and *Nlrc4^–/–^* OvaFla mice and pulsed the mice with a single day of tamoxifen chow (our standard protocol). We harvested spleens and mesenteric lymph nodes five days later. We were able to recover a small population of congenically-marked OT-IIs in the mice; however, these cells were not dividing (based on CTV labeling) and remained native (CD62L^+^ CD44^–^) (Author response image 1). We repeated this experiment and obtained the same results.

**Author response image 1. sa2fig1:** Adoptively transferred OT-II T cells fail to proliferate or become CD62L^–^CD44^+^ in OvaFla mice by day five post tamoxifen chow pulse. Shown are representative flow plots demonstrating the gating strategy for identifying OT-Iis in the mesenteric lymph nodes of WT (top), *Gsdmd^–/–^* (middle), or *Nlrc4^–/–^* (bottom) OvaFla mice.

Unfortunately, it is difficult to determine whether the lack of an OT-II response is because NAIP–NLRC4 activation is not sufficient to activate OVA-specific CD4^+^ T cells or because the OT-IIs are less sensitive to small amounts of MHC-presented OVA peptide than OT-Is. Figure 1 in Li M. *et al.,* 2001 *J Immunol* (PMID 11342628) shows that, when compared with OT-Is, OT-IIs require ten-fold more OVA peptide before they show robust division in vivo, so we speculate that limiting antigen levels may be the problem here.

2. The authors should comment on whether endogenous OVA-specific CD8^+^ T cell responses can be detected in these models (e.g. using tetramers or ELISPOT assays).

We used two separate approaches to identify endogenous OVA-specific CD8^+^ T cells responses in the WT, *Gsdmd^–/–^*, and *Nlrc4^–/–^* OvaFla mouse lines. For the first approach, we pulsed the mice with one day of tamoxifen chow and then harvested mesenteric lymph nodes seven days later. We stained each sample with a SIINFEKL-MHC Class I tetramer and used magnetic enrichment to enhance our ability to find potentially rare OVA-specific CD8^+^ T cell populations (PMID 31034767) (Author response image 2). We confirmed reactivity of the tetramer with mice given OT-Is as a positive control; however, we were unable to detect tetramer positive CD8^+^ T cells in any of the experimental groups. We repeated this experiment in chimeric K^bm1+^ OvaFla mice, and we were again unable to identify a clear tetramer-positive population among endogenous T cells.

**Author response image 2. sa2fig2:** Endogenous SIINFEKL-specific CD8^+^ T cells cannot be found using tetramer staining in OvaFla mice at seven days post tamoxifen chow pulse. Shown are representative flow plots demonstrating the gating strategy used to identify CD8^+^ T cells that recognize SIINFEKL presented on MHC I. Included is a positive control from a mouse given OT-I T cells and a negative control naïve B6 mouse.

For the second approach, we followed the same tamoxifen time course outlined above, but we used an IFNγ ELISpot assay to detect CD8^+^ T cells that are responding to SIINFEKL peptide at seven days post tamoxifen chow pulse. We were again unable to detect a peptide-specific response in endogenous CD8^+^ T cells.

The lack of an endogenous response is likely due to technical limitations of our system. As is now more clearly noted in the manuscript starting on line 245, we occasionally see some evidence for (mild) OT-I activation in OvaFla mice lacking the Cre recombinase (Figure 3—figure supplement 1D, 1E). This suggests that the OvaFla transgene might be expressed at low levels in a Cre-independent manner. Chronic expression of OvaFla as a self-antigen would likely result in the exhaustion or deletion of the endogenous SIINFEKL-specific CD8^+^ T cells, explaining why we find it difficult to see an endogenous response. Our strategy of transferring naïve OT-I T cells from an OvaFla-negative host circumvents this problem. Essentially, though, we believe that our system is useful primarily for studying the fate and presentation of IEC-derived antigen, using acute responses of OT-I T cells as a sensitive detector of this antigen, but is not useful for studying the response or downstream effector function of endogenous cells. This limitation is also relevant to point 4 below, and we believe it is an important point that we now discuss in the manuscript (lines 253-257). While it is possible that NAIP–NLRC4 activation in IECs is not sufficient to induce an endogenous CD8^+^ T cell response to IEC-derived antigens, we do not feel that it is appropriate to draw that conclusion from our current negative data.

3. The differential expression of OVA, assessed by the GFP reporter in Figure 2, is concerning. The authors have a 2-variable system here, with mice contain different amounts of antigen and different pyroptosis-inducing potential. The authors acknowledge this point on line 216, but never do additional work to accurately quantify GFP expression across the genotypes of mice they are focused on. Additional quantification of transgene expression is critical, as the phenotypes described later in the study are modest.

The reviewer raises a key issue that we tried to highlight in the initial version, though it is agreed further analysis is in order. We fully agree that the differential expression of OVA between the WT and *Nlrc4*^–/–^ OvaFla mice adds a second variable to our OvaFla system that is important to control for. In the revised manuscript, we now include new quantification of GFP expression in the IECs of WT, *Nlrc4^–/–^* and *Gsdmd^–/–^* OvaFla mice in Figure 2C. Importantly, it appears that the OvaFla-GFP transgene is expressed at indistinguishable levels in WT and *Gsdmd*^–/–^ intestinal epithelial cells. Because the levels are indistinguishable, we believe the comparison of OVA-specific CD8^+^ T cell activation between these mice can be attributed to whether or not the IECs are able to undergo pyroptosis following NAIP–NLRC4 expression.

4. Monitoring T cell response at day 5 post-treatment does not allow the authors to assess CD8 T cell differentiation into memory cells or functions in the context of infectious challenges with pathogens that express the OVA transgene. A suitable revision would include such analysis, where the authors examine later time points for T cell functions (20-40 days, for example) and the authors challenge these mice with pathogens (Listeria-OVA, for example), to assess functional CTL responses in a relevant setting.

As discussed above in response to point 2, we agree that our manuscript focuses only on the early events of T cell activation after NAIP–NLRC4 activation and pyroptosis. We agree with the reviewer that exploring later effects is of interest, though (as discussed above) there are technical difficulties in trying to do this with our OvaFla system. Nevertheless, in an attempt to understand how NAIP–NLRC4 activation and pyroptosis might influence the recall response of these CD8^+^ T cells, we adoptively transferred OT-Is into WT, *Gsdmd^–/–^*, and *Nlrc4^–/–^* OvaFla mice, pulsed these mice with one day of tamoxifen chow, and then let the animals rest for >30 days. We then infected these mice with 1×10^5^ CFU of *Listeria*-OVA. We monitored the mice for weight loss and harvested the spleen and liver on day 5 post-infection. Unfortunately, we were unable to recover any of our transferred OT-Is. We believe that the chronic expression of the OvaFla transgene that likely occurs even after withdrawal of tamoxifen (PMID 15282745) results in tolerization and/or deletion of the OT-I T cells. In addition, we found that the *Nlrc4^–/–^* OvaFla mice had a significantly higher CFU in their spleens relative to the other mouse lines, which further complicates the interpretation of any experiment (even if we were able to figure out how to track CD8^+^ cells long term in our system).

Nevertheless, as an alternative approach, we tried an adoptive transfer approach. In this approach we again transferred OVA-specific CD8^+^ T cells into each of the OvaFla mouse lines and gave these mice a single day pulse of tamoxifen chow. This time, however, we adoptively transferred the OT-Is from these OvaFla mice into naïve B6 mice. These B6 mice were rested for >30 days and then challenged with 1×10^5^ CFU of *Listeria*-OVA. Unfortunately, we were again unable to recover OT-Is in any mice across two separate experiments.

From the above experiments, it is difficult to determine if our inability to find a memory CD8^+^ T cell response to *Listeria*-OVA after OvaFla induction is due to technical difficulties or a biological reason. Because Villin-driven Cre recombinase is capable of recombining in the stem cell laden intestinal crypts, which continuously produce nascent IECs (PMID 15282745), we hypothesized that the OvaFla might become a chronic self antigenic stimulus, leading to CD8^+^ T cell exhaustion. To test this hypothesis, we looked at PD-1 expression on our adoptively transferred OT-Is, as this is a marker of chronic antigen exposure (e.g., see Youngblood B, *et al.*, 2011 *Immunity* PMID 21943489). We collected spleens and mesenteric lymph nodes at days 11 and 14 post initial tamoxifen chow pulse and OT-I transfer (Author response image 3). Interestingly, we found that nearly all of the remaining OT-Is in the mesenteric lymph nodes of the *Nlrc4^–/–^*OvaFla mice were PD-1^+^, suggesting that these cells have become exhausted. We conclude that while the OvaFla system is a powerful genetic system to examine the effect of inflammasome activation on antigen presentation and early events during T cell activation, technical limitations (primarily an inability to turn off OvaFla expression) prevent us from studying memory responses using this system.

**Author response image 3. sa2fig3:** OT-Is in WT, *Gsdmd^–/–^*, and *Nlrc4^–/–^* OvaFla mice upregulate PD-1 at 11 and 14 days after tamoxifen chow pulse. A. Total number of OT-Is in the mesenteric lymph nodes (left) and spleens (right) of the indicated OvaFla mouse lines at 11 and 14 days after tamoxifen chow pulse. B. Percent of OT-Is in the mesenteric lymph nodes (left) and spleens (Right) of the indicated OvaFla mouse lines that are expressing PD-1 at 11 and 14 days after tamoxifen chow pulse. Data are pooled from two biological replicates, and each dot represents an individual mouse. Data shown as mean ­± SD. Significance calculated using two-way ANOVA and Tukey’s multiple comparisons test (**p* < 0.05, ***p* < 0.01, ****p* < 0.001).

5. While the OVA Fla+Villin ER-Cre model is elegant, it would have been better if the authors had also created a model to directly compare the steady state cross-presentation to CD8 T cells. The NLRC4 KO data suggest that there is steady state cross presentation happening in the absence of NLRC4 activation and there is no increase in cross-presentation because of NLRC4 activation and subsequent pyroptosis. Is it possible that the quality of primed CD8 T cells is different in the presence and absence of NLRC4. While the author examined IFNγ TNF production, granzyme B, Perforin expression or pathogen clearance experiments (Listeria OVA for example) would have tested the quality of primed CD8 T cells.

We agree that the NLRC4 KO OvaFla mice do provide a system for studying steady-state (non-inflammatory) cross-presentation of IEC-derived antigens. We also agree that a pathogen challenge experiment to compare the quality of primed OVA-specific CD8^+^ T cells in the presence and absence of NLRC4 activation would add to our understanding of the effects of inflammasome activation on cross presentation. However, per our response to Essential Revision #4, we have been unsuccessful in our attempts to measure T cell responses after challenging the OvaFla mice with *Listeria*–OVA. Additionally, we have been unable to achieve clear staining for granzyme B or perforin in our activated OT-I CD8^+^ T cells. However, our results do nevertheless uncover an important distinction between the presentation of antigens in NLRC4-deficient vs NLRC4-sufficient mice. In particular, though the use of bone marrow chimeras with *Batf3^–/–^* (Figure 5) and *Zbtb46^–/–^* (Figure 6, new data) donors, we discovered that cross presentation of IEC antigen following NAIP–NLRC4 activation occurs via a separate population of cDCs (presumably cDC2s) relative to steady state cross presentation in the NLRC4-deficient mice (see Discussion in manuscript and in point 7 below).

6. OT-I cross priming being independent of NLRC4 but partially dependent on gasdermin D is very interesting. Did the authors examine the MHC Class I-SIINFEKL complexes on DCs in the mesenteric lymph nodes by using either a clonotypic antibody (clone 25-D1.16) or in vitro priming of OT-I T cells by isolating DCs from mLNs. Similarly, was the activation status of cDC1s different between WT, Gasdermin DKO and NLRC4 KO mice after tamoxifen administration?

To examine DC presentation of the SIINFEKL peptide in the OvaFla mice, we used the highly sensitive B3Z assay to quantify the amount of SIINFEKL presented on MHC Class-I (PMID: 1378619). The B3Z assay uses a T cell hybridoma cell line, where TCR recognition of the MHC Class I-SIINFEKL complex activates a lacZ reporter. WT, *Gsdmd^–/–^*, and *Nlrc4^–/–^* OvaFla mice were given two days of tamoxifen chow, and on the second day, we collected mesenteric lymph nodes and depleted lymphocytes from the single cell suspensions. While the assay technically worked, as indicated by a positive control in which SIINFEKL peptide was added to some of the cells from the mesenteric lymph nodes, we were unable to detect any endogenous SIINFEKL presentation in two separate experiments. Even though the B3Z assay is designed to detect rare APCs, we suspect the frequency of SIINFEKL-MHC Class I DCs is too low in our system to accurately quantify ex vivo.

As suggested by the reviewers, to examine the activation status of cDC1s across the WT, *Gsdmd^–/–^*, and *Nlrc4^–/–^* OvaFla mice, we performed a similar experiment, where we gave mice from each of these lines two days of tamoxifen chow. After the third day, we harvested the spleens and lymph nodes and looked for evidence of DC maturation using the upregulation of MHC II and CD86 (also known as B7-2). We did not see any difference in cDC maturation across the OvaFla mouse lines and have added these data to Figure 5—figure supplement 1 in the revised manuscript. We suspect that the low numbers of DCs that respond and are activated in this system are not detectable by flow cytometry, and therefore the CD8^+^ T cell response remains our most sensitive reporter of antigen presentation.

7. The study does point out that there is a Batf3 independent pathway of antigen cross -presentation to T cells that might be occurring upon activation of NLRC4 inflammasome however, the study overall might benefit from a better explanation of what other pathways or other populations of DCs might be facilitating the antigen presentation in this case. The authors do mention the recent study by Bosteel et al., briefly in the discussion but characterization of the inflammatory cDC2 population in the Nlrc4-/- might provide a better insight into the mechanism of antigen presentation.

We agree that our manuscript would benefit from better characterization of the *Batf3*-independent cross-presentation pathway seen in the WT OvaFla mice. Because red pulp macrophages (PMID: 31776205) and monocyte-derived DCs (PMID: 27264183) were recently shown to cross-present antigens in a *Batf3*-independent manner, we wanted to determine if the *Batf3-*independent APC population we see is another cDC population—presumably cDC2s—or a macrophage or moDC population.

For these experiments, we generated bone marrow chimeras where the bm1^+^ OvaFla mice were lethally irradiated and reconstituted with bone marrow from either B6 mice or *Zbtb46*-diptheria toxin receptor (DTR) mice (Jax strain: 019506). The transcription factor *Zbtb46* drives development of classical DCs (cDC1s and cDC2s) but not other myeloid cells, including macrophages and moDCs (PMID: 22615127). Eight weeks after irradiation and reconstitution, we treated all chimeric mice with DT one day prior to OT-I transfer and tamoxifen chow pulse. The mice were treated again with DT two days after the OT-I transfer, and we euthanized the mice and collected spleens and mesenteric lymph nodes at day five post transfer. In contrast to the previous experiments using bone marrow from *Batf3*-deficient mice, none of the OvaFla lines that received bone marrow from *Zbtb46*-deficient mice had robust OT-I populations in their secondary lymphoid organs. These data suggest that the *Batf3*-independent cross-presenting population found following NAIP–NLRC4 activation in IECs requires *Zbtb46* (and is thus likely due to cDC2s). We believe these results add considerably to the manuscript, and we have updated the manuscript to include these findings in the section titled “cDCs are required for cross-presentation of IEC derived antigen” (beginning on line 378) and Figure 6.

8. While the study sets out to investigate the role of inflammasome activation and pyroptosis in CD8 T cell cross priming, based on the results, a more appropriate title would be" Inflammasome activation leads to cDC1 independent cross-priming of CD8 T cells by epithelial cell derived antigen".

This is a terrific suggestion with which we fully agree. We have changed the title of the resubmitted manuscript to reflect this very helpful suggestion.